# Iterative Robust Satisficing: Minimizing Performance Degradation Under Distribution Shift

Enes Ağırman [* 1]    Artun Saday [* 1]    Cem Tekin [1]

## Abstract

Modern neural networks often achieve high accuracy on their training distribution but degrade sharply under distribution shifts. We address this problem through *Robust Satisficing* (RS), an optimization objective that seeks parameters which attain a target level of in-distribution performance while minimizing *fragility*, defined as the rate at which performance deteriorates as the data distribution departs from training. We develop a gradient-based algorithm, *Iterative Robust Satisficing* (IRS), that directly optimizes this criterion. Across a range of synthetic and real-world distribution shifts, including long-tailed image classification, group shifts induced by spurious correlations, and natural shifts in tabular regression, IRS consistently improves performance on minority and worst-case groups without sacrificing overall accuracy. Notably, IRS achieves these robustness gains with a per-step computational cost similar to standard stochastic gradient descent and requires only a single forward and backward pass per update. Together, these results suggest that minimizing fragility provides a practical and effective alternative to existing robust training methods for learning models that remain reliable under distribution shift.

## 1. Introduction

Developing models that remain reliable under changing deployment conditions is essential for real-world use (Tran et al., 2025). Yet modern neural networks trained with Empirical Risk Minimization (ERM) can be highly sensitive to distribution shifts, performance often degrades sharply even under seemingly minor changes to the data-generating process (Recht et al., 2019; Taori et al., 2020; Chen et al., 2023). This vulnerability is especially problematic in deep neural networks, where high capacity can worsen overfitting to the training distribution while leaving the model fragile under distributional perturbations (Li et al., 2022).

A variety of approaches have been proposed to improve the out-of-distribution (OOD) generalization of neural networks (NNs). Early work on robust optimization in machine learning drew on the min–max tradition in operations research, formulating objectives that guard against worst-case losses within an uncertainty set of distributions (Ben-Tal et al., 2009; Bertsimas et al., 2011). One prominent line of work is distributionally robust optimization (DRO), which seeks to minimize the worst-case loss over an uncertainty set of data distributions. Although DRO directly targets distribution shifts, its worst-case nature can be pessimistic, and tail performance may come at the expense of in-distribution performance (Duchi & Namkoong, 2021). Several DRO formulations and scalable algorithms have been developed, including $f$-divergence DRO with statistical guarantees (Namkoong & Duchi, 2016), Wasserstein DRO for geometric shifts (Kuhn et al., 2019; Gao & Kleywegt, 2023), adversarial-training inspired formulations (Sinha et al., 2018), and large-scale methods that make DRO practical in modern deep learning pipelines (Levy et al., 2020). For group-structured shifts, GroupDRO explicitly optimizes worst-group performance and has become a standard baseline for spurious correlations and subpopulation robustness (Sagawa et al., 2020).

Beyond DRO, robustness has been pursued through alternative principles. Sharpness-Aware Minimization (SAM) seeks parameters in flat regions of the training loss landscape, motivated by the empirical link between sharpness and generalization (Foret et al., 2021). Invariance-based approaches aim to learn predictors that remain stable across multiple training environments, including invariant risk minimization (IRM) (Arjovsky et al., 2019) and related representation learning methods (Ganin et al., 2016). Risk extrapolation (REx) methods such as V-REx and MM-REx (Krueger et al., 2021) regularize dispersion of risks across environments to improve OOD generalization. While effective in some regimes, many of these approaches require environment/group annotations or rely on assumptions that

---

[*]Equal contribution  [1]Department of Electrical and Electronics Engineering, Bilkent University. Correspondence to: Enes Ağırman <enes.agirman@bilkent.edu>.

*Proceedings of the 43rd International Conference on Machine Learning*, Seoul, South Korea. PMLR 306, 2026. Copyright 2026 by the author(s).

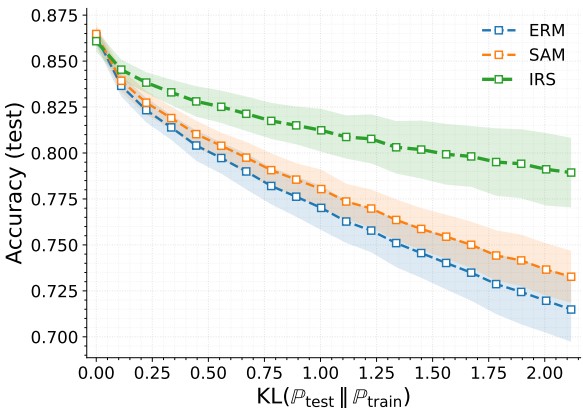

*Figure 1.* Accuracy under increasing distribution shift (measured by the KL divergence between the training and test distributions). As the shift grows, ERM and SAM degrade sharply, while IRS remains substantially more stable. Means over 100 runs; shaded regions show $\pm 1$ standard deviation.

may not hold in practice (Rosenfeld et al., 2021; Schölkopf et al., 2021; Li et al., 2018).

Recently, *Robust Satisficing* (RS) has emerged as an alternative to worst-case DRO for handling distributional uncertainty. Rather than minimizing worst-case risk, RS seeks parameters that (i) satisfy a target in-distribution performance level and (ii) minimize *fragility*, the rate at which performance degrades as the data distribution departs from the training distribution. This perspective has roots in decision theory (Schwartz et al., 2011) and was formalized as an optimization objective by Long et al. (2023). RS has since been studied theoretically (Li et al., 2024) and applied in several domains, including Bayesian optimization (Saday et al., 2023; 2025) and robust decision-making in Markov decision processes (Ruan et al., 2023). Most recently, Yan et al. (2024) introduced a KL-divergence–based RS formulation for neural network training (KL-RS). However, KL-RS estimates fragility via a threshold search procedure that retrains the network multiple times, substantially increasing computational cost and limiting scalability to large models and datasets. Our goal is to develop an efficient and robust procedure for training models under the RS objective. Table 1 positions IRS relative to other known optimization objectives.

**Contributions**  We propose *Iterative Robust Satisficing* (IRS), a gradient-based procedure for training neural networks under the RS objective. Our key technical insight is that the inner maximization in the objective can be reduced from a search over the probability simplex to a one-dimensional scalar trajectory, enabling efficient optimization. As a result, IRS optimizes the RS objective within a *single* training run and incurs nearly the same per-iteration cost as standard SGD (one forward and one backward pass)

up to a low-cost scalar search. Empirically, across synthetic and real distribution shifts, IRS consistently improves robustness and worst-group performance relative to strong baselines, while preserving in-distribution accuracy (e.g., Figure 1).

## 2. Problem Formulation

We denote scalars by lowercase letters ($x \in \mathbb{R}$), vectors by bold lowercase letters ($\boldsymbol{x} \in \mathbb{R}^p$), and matrices by bold uppercase letters ($\boldsymbol{X} \in \mathbb{R}^{p \times n}$). Sets are denoted by calligraphic letters (e.g., $\mathcal{X}$), and probability distributions by script letters (e.g., $\mathbb{P}, \mathbb{Q}$). In our setting, let $p$ denote the input feature dimension, $n$ the number of training samples, and $d$ the number of learnable model parameters. A summary of notation is provided in Appendix A. For a model parameterized by $\theta \in \Theta \subseteq \mathbb{R}^d$ (with $\Theta = \mathbb{R}^d$ in our neural-network setting), we define the expected loss under a distribution $\mathbb{P}$ as $L_{\mathbb{P}}(\theta) := \mathbb{E}_{(x,y) \sim \mathbb{P}} [ \ell(\theta, x, y) ]$, where $\ell : \Theta \times \mathcal{X} \times \mathcal{Y} \to \mathbb{R}_+$ is the per-sample loss function. Also let $\mathcal{P}_0$ denote the set of all distributions on $\mathcal{X} \times \mathcal{Y}$. The empirical distribution associated with a dataset $\mathcal{D} = \{z_i\}_{i=1}^n = \{(x_i, y_i)\}_{i=1}^n$ is denoted by $\widehat{\mathbb{P}}_{\mathcal{D}} := \frac{1}{n} \sum_{i=1}^n \delta_{(x_i, y_i)}$, where $\delta_{(x_i, y_i)}$ denotes the Dirac measure at point $(x_i, y_i)$. Let $\Delta^{n-1} := \{ \boldsymbol{p} \in \mathbb{R}_+^n : \mathbf{1}^\top \boldsymbol{p} = 1 \}$ denote the probability simplex over $n$ training samples.

To reason about distribution shifts, we introduce a probability divergence $d : \mathcal{P}_0 \times \mathcal{P}_0 \to \mathbb{R}_+$ that quantifies the distance between two probability measures $\mathbb{P}$ and $\mathbb{Q}$ on $(\mathcal{X}, \mathcal{Y})$. Common choices include the Wasserstein distance $d_{\mathrm{W}}(\mathbb{P}, \mathbb{Q}) := \inf_{\pi \in \Pi(\mathbb{P}, \mathbb{Q})} \mathbb{E}_{(x,y),(x',y') \sim \pi} [\|(x, y) - (x', y')\|]$, where $\Pi(\mathbb{P}, \mathbb{Q})$ denotes the set of all joint distributions on $(x, y), (x', y')$ with marginals $\mathbb{P}$ and $\mathbb{Q}$, and $f$-divergences such as the Kullback–Leibler divergence $d_{\mathrm{KL}}(\mathbb{P} \| \mathbb{Q}) := \mathbb{E}_{(x,y) \sim \mathbb{P}} [\log \frac{d\mathbb{P}}{d\mathbb{Q}}(x, y)]$.

We are concerned with the gradient-based optimization of a trainable model. The ERM approach aims to directly minimize the expected loss with respect to the training distribution $\widehat{\mathbb{P}}_{\mathcal{D}}$ by solving,

$$\theta^{\mathrm{ERM}} \in \arg\min_{\theta \in \Theta} L_{\widehat{\mathbb{P}}_{\mathcal{D}}}(\theta) . \tag{1}$$

Let $\mathbb{P}_d$ denote the deployment distribution. When $\mathbb{P}_d$ differs from the training distribution, the solution of (1) does not provide strong performance guarantees. One approach to tackle this problem is known as distributionally robust optimization (DRO) which aims to minimize the worst case loss over an ambiguity set of possible distributions by solving,

$$\theta^{\mathrm{DRO}} \in \arg\min_{\theta \in \Theta} \max_{\mathbb{P} \in \mathcal{U}} L_{\mathbb{P}}(\theta) , \tag{2}$$

where $\mathcal{U}$ is an ambiguity set of candidate test distributions, typically defined as $\mathcal{U}(\rho) := \{ \mathbb{Q} \in \mathcal{P}_0 \mid d(\mathbb{Q}, \widehat{\mathbb{P}}_{\mathcal{D}}) \leq \rho \}$,

| Method | Objective | Cost / Requirements |
|---|---|---|
| ERM | $\min_{\theta \in \Theta} \ \mathbb{E}_{\widehat{\mathbb{P}}_{\mathcal{D}}}[\ell(\theta, x, y)]$ | 1F + 1B |
| SAM | $\min_{\theta \in \Theta} \ \max_{\|\xi\|_p \leq \rho} \ \mathbb{E}_{\widehat{\mathbb{P}}_{\mathcal{D}}}[\ell(\theta + \xi, x, y)]$ | 2F + 2B |
| DRO | $\min_{\theta \in \Theta} \ \max_{\mathbb{P} \in \mathcal{U}} \ \mathbb{E}_{\mathbb{P}}[\ell(\theta, x, y)]$ | Algorithm dependent |
| IRM | $\min\limits_{\theta \in \Theta} \sum\limits_{e \in \mathcal{E}_{\text{tr}}} \left[ R^e(\theta) + \lambda \left\| \nabla_{w \,|\, w=1} R^e(w \cdot f_\theta) \right\|_2^2 \right]$ | 1F + 1B + higher-order grad. + env. labels |
| KL-RS | $\min_{\theta \in \Theta} \ \kappa_\tau(\theta)$ | Multiple full trainings |
| **IRS (ours)** | $\min_{\theta \in \Theta} \ \kappa_\tau(\theta)$ | 1F + 1B + scalar optimization |

*Table 1.* F and B denote forward and backward passes per model-parameter update. $e \in \mathcal{E}_{\text{tr}}$ indexes training environments, $f_\theta$ denotes the predictor parameterized by $\theta$, $R^e(\theta)$ the risk in environment $e$, $w$ is the fixed "dummy" classifier, and $\lambda$ is the penalty weight used in IRM (Arjovsky et al., 2019). $\xi$ denotes the SAM weight perturbation with radius $\rho$. The table summarizes representative optimization objectives; additional robustness baselines, including GroupDRO, V-REx, and MM-REx, are evaluated in the experiments. KL-RS requires multiple full trainings for different thresholds, whereas IRS optimizes the same objective in a single run with an additional low-cost scalar optimization.

with $\rho > 0$ the radius controlling the size of the set. Defining a proper ambiguity set can be difficult when $d(\mathbb{P}_d, \widehat{\mathbb{P}}_{\mathcal{D}})$ cannot be estimated precisely. We propose another way that approaches model training from the lens of RS, whose mathematical characterization is first proposed in (Long et al., 2023). In RS, instead of minimizing the ERM, the goal is to find a solution $\theta^{\text{RS}} \in \Theta$, such that the model achieves performance that at least matches a target level $\tau$ under the empirical distribution, that is $L_{\widehat{\mathbb{P}}_{\mathcal{D}}}(\theta^{\text{RS}}) \leq \tau$. When $\mathbb{P}_d \neq \widehat{\mathbb{P}}_{\mathcal{D}}$, the shortfall in performance with respect to $\tau$ should increase at most linearly and in the smallest possible way in $d(\mathbb{P}_d, \widehat{\mathbb{P}}_{\mathcal{D}})$. In other words, the rate of shortfall in performance, also called fragility, should be minimized. The fragility of a given parameter $\theta$ is formally defined as

$$\kappa_\tau(\theta) = \min k \ \text{ subject to } \ L_{\mathbb{P}}(\theta) \leq \tau + k \, d(\mathbb{P}, \widehat{\mathbb{P}}_{\mathcal{D}}),$$
$$\forall \mathbb{P} \in \mathcal{P}_0, \ k \geq 0. \tag{3}$$

Equivalently, $\kappa_\tau(\theta)$ is the smallest slope of a linear upper envelope certifying $L_{\mathbb{P}}(\theta) \leq \tau + \kappa_\tau(\theta) d(\mathbb{P}, \widehat{\mathbb{P}}_{\mathcal{D}})$ for every candidate distribution $\mathbb{P} \in \mathcal{P}_0$; distributions closer to $\widehat{\mathbb{P}}_{\mathcal{D}}$ therefore receive tighter loss certificates. The RS solution is the one with the smallest fragility, $\theta^{\text{RS}} \in \arg \min_\theta \kappa_\tau(\theta)$. There are two distinct characteristics of fragility that sets it apart from the well known robust objective DRO.

- Firstly, the RS solution $\theta^{\text{RS}}$ has to achieve a set performance threshold $\tau$ under the empirical distribution. It is known that DRO solutions can suffer from being overly conservative and performing poorly under the empirical distribution (Duchi et al., 2021).

- Secondly, the DRO solution only gives performance guarantees for distributions that are in the given ambiguity set. If the deployment distribution lies outside this ambiguity set, DRO fails to give any performance guarantees. In contrast, the fragility formulation in (3) is defined against arbitrary deployment distributions

in $\mathcal{P}_0$ and, in particular, does not require the modeler to commit to a radius around $\widehat{\mathbb{P}}_{\mathcal{D}}$. Section 3 describes the standard empirical approximation used by our algorithm, in which $\mathcal{P}_0$ is restricted to reweightings of the observed support.

## 3. Implementation of IRS

In practice our algorithm operates in the region where $L_{\widehat{\mathbb{P}}_{\mathcal{D}}}(\theta) < \tau$ and $\max_i \ell(\theta, z_i) > \tau$, i.e. the in-distribution loss is below the satisficing level but some atom violates it. In this regime, which is our main focus in this section, the fragility can be written as

$$\kappa_\tau(\theta) = \max_{\mathbb{P} \in \mathcal{P}_0 \setminus \{\widehat{\mathbb{P}}_{\mathcal{D}}\}} \frac{L_{\mathbb{P}}(\theta) - \tau}{d(\mathbb{P}, \widehat{\mathbb{P}}_{\mathcal{D}})}. \tag{4}$$

The other regimes are summarized as follows: if $L_{\widehat{\mathbb{P}}_{\mathcal{D}}}(\theta) > \tau$ the satisficing target is infeasible and $\kappa_\tau(\theta) = \infty$; if $\max_i \ell(\theta, z_i) \leq \tau$ then $\kappa_\tau(\theta) = 0$ and the gradient step in Algorithm 1 is skipped. The boundary case $L_{\widehat{\mathbb{P}}_{\mathcal{D}}}(\theta) = \tau$ and a formal derivation of (4) are given in Appendix D.1.

Our aim is to optimize the model parameters not directly through the loss but through their fragility, using a gradient-based approach. Note, however, that $\nabla_\theta \kappa_\tau(\theta)$ is well defined only when $L_{\widehat{\mathbb{P}}_{\mathcal{D}}}(\theta) < \tau$, a condition rarely satisfied when training a neural network from an arbitrary initialization. To address this, we adopt a *threshold-scheduling strategy*. At each optimization step $t$, we set an adaptive threshold $\tau_t := (1 + \epsilon) L_{\widehat{\mathbb{P}}_{\mathcal{D}}}(\theta_t)$, with $\epsilon > 0$, compute the fragility $\kappa_{\tau_t}(\theta_t)$ of the parameter $\theta_t$ at time $t$, and its gradient, and update the parameters according to

$$\theta_{t+1} = \theta_t - \eta \, \nabla_\theta \kappa_{\tau_t}(\theta_t), \tag{5}$$

where $\eta > 0$ is the learning rate.[1] As training progresses, the empirical loss $L_{\widehat{\mathbb{P}}_{\mathcal{D}}}(\theta_t)$ decreases, causing the scheduled

---

[1]We use the usual unconstrained neural-network parameteriza-

threshold $\tau_t$ to shrink. Once $L_{\widehat{\mathbb{P}}_{\mathcal{D}}}(\theta_t)$ drops below the target $\tau$, we replace $\tau_t$ with the original threshold $\tau$ and continue optimizing the fragility with respect to that fixed value.

**Standing assumptions.** Throughout Section 3 we assume:

(A1) For each $(x, y)$, the per-sample loss $\ell(\theta, x, y)$ is continuously differentiable in $\theta$.

(A2) For every $\theta$ considered, the inner maximization in (4) attains a unique maximizer $\boldsymbol{p}^*(\theta) \in \mathcal{P}_0$.

Under (A1)–(A2), the simple $C^1$ form of Danskin's theorem (in Appendix D.3) applies to the inner problem in (4).

### 3.1. Computing the Gradient

The population definition in Section 2 quantifies fragility against arbitrary distributions on $\mathcal{X} \times \mathcal{Y}$. For algorithmic tractability, we adopt the standard empirical approximation and restrict our attention to *atomic data distributions*: we assume that $\mathcal{P}_0$ consists of all probability measures supported on the finite set $\mathcal{Z} = \{z_1, \ldots, z_n\}$ of observed samples, with $z_i = (x_i, y_i)$. Equivalently, $\mathcal{P}_0$ is the probability simplex

$$\mathcal{P}_0 := \Big\{ p \in \mathbb{R}^n_+ \ \big| \ \sum_{i=1}^{n} p_i = 1 \Big\},$$

so each $\mathbb{P} \in \mathcal{P}_0$ can be represented as the vector $\boldsymbol{p} = [p_1, \ldots, p_n]^\top$ of weights assigned to the atoms. Under this approximation the inner maximization in (3) ranges over reweightings of the observed support; the infinite-dimensional optimization over distributions becomes a finite-dimensional problem on the simplex, and the resulting objective is the standard empirical counterpart of the population formulation.

In order to compute $\nabla_\theta \kappa_\tau(\theta)$, we first need to solve (4). For the moment, we assume access to an oracle that returns the distribution attaining the maximum in (4). Denote this maximizer by

$$\mathbb{P}^*(\theta) \in \operatorname*{arg\,max}_{\mathbb{P} \in \mathcal{P}_0 \setminus \{\widehat{\mathbb{P}}_{\mathcal{D}}\}} \frac{L_{\mathbb{P}}(\theta) - \tau}{d\big(\mathbb{P}, \widehat{\mathbb{P}}_{\mathcal{D}}\big)},$$

which gives

$$\kappa_\tau(\theta) = \frac{L_{\mathbb{P}^*}(\theta) - \tau}{d\big(\mathbb{P}^*, \widehat{\mathbb{P}}_{\mathcal{D}}\big)}.$$

Later, we will present an efficient algorithm to calculate $\mathbb{P}^*(\theta)$, replacing the oracle. Having expressed $\kappa_\tau(\theta)$

---

tion, $\Theta = \mathbb{R}^d$, so the update in (5) is an unconstrained gradient step. If an external constraint on $\theta$ is imposed, the step can be composed with the appropriate projection onto $\Theta$. This affects only the outer parameter update, not the inner $\boldsymbol{p}$-maximization or the Danskin gradient formula.

solely as a function of the parameter $\theta$, we can now differentiate it with respect to $\theta$ and derive the corresponding gradient update rule. Define the probability vectors $\boldsymbol{p}^* = \mathbb{P}^*(\theta)$ and $\hat{\boldsymbol{p}} = \widehat{\mathbb{P}}_{\mathcal{D}}$, and the per-sample loss vector $\boldsymbol{\ell}_\theta = [\ell(\theta, x_i, y_i)]_{i=1}^{n}$. Note that while $\boldsymbol{p}^*$ and $\boldsymbol{\ell}_\theta$ are functions of $\theta$, $\hat{\boldsymbol{p}}$ only depends on the data. Then the expected loss under the $\mathbb{P}^*$ is $L_{\mathbb{P}^*}(\theta) = \boldsymbol{p}^{*\top} \boldsymbol{\ell}_\theta$.

**Proposition 3.1** (Gradient of $\kappa_\tau$)**.** *The gradient of $\kappa_\tau(\theta)$ defined in* (4) *is initially given by*

$$\nabla_\theta \kappa_\tau(\theta) = \frac{\nabla_\theta[\boldsymbol{p}^{*\top}\boldsymbol{\ell}_\theta]\, d(\boldsymbol{p}^*, \hat{\boldsymbol{p}}) + (\tau - \boldsymbol{p}^{*\top}\boldsymbol{\ell}_\theta)\nabla_\theta d(\boldsymbol{p}^*, \hat{\boldsymbol{p}})}{d(\boldsymbol{p}^*, \hat{\boldsymbol{p}})^2} \tag{6}$$

*By Danskin's theorem, the derivative of the maximized value is obtained by taking the partial derivative with respect to $\theta$ at the active maximizer $\boldsymbol{p}^*(\theta)$; no derivative through the argmax map is needed. Thus, in this partial derivative, $\boldsymbol{p}^*$ is held fixed and $d(\boldsymbol{p}^*, \hat{\boldsymbol{p}})$ has no remaining $\theta$-dependence. Substituting these simplifications into* (6) *yields*

$$\nabla_\theta \kappa_\tau(\theta) = \frac{\boldsymbol{p}^{*\top}\nabla_\theta \boldsymbol{\ell}_\theta}{d(\boldsymbol{p}^*, \hat{\boldsymbol{p}})}. \tag{7}$$

*Proof.* We get (6) by the quotient rule, and Danskin's theorem implies that when differentiating the max value with respect to $\theta$, the partial derivative is evaluated at the active maximizer $\boldsymbol{p}^*(\theta)$ without differentiating through the argmax map. Hence $\boldsymbol{p}^*$ is held fixed in this derivative, and $\nabla_\theta d(\boldsymbol{p}^*, \hat{\boldsymbol{p}}) = \boldsymbol{0}$. Thus $\nabla_\theta[\boldsymbol{p}^{*\top}\boldsymbol{\ell}_\theta] = \boldsymbol{p}^{*\top}\nabla_\theta\boldsymbol{\ell}_\theta$, yielding (7). $\square$

The gradient in (7) can be readily computed using modern automatic differentiation libraries such as PyTorch, TensorFlow, or JAX.

### 3.2. Finding $\kappa_\tau(\theta)$ via $\boldsymbol{p}^*$

To evaluate the gradient in (7), we must identify the distribution $\boldsymbol{p}^*$ that attains the maximum in (4). However, solving (4) exactly is challenging because the solution is constrained to lie on the probability simplex $\mathcal{P}_0$. To make the computation tractable, we adopt a Lagrangian dual approach. By definition

$$\sup_{\boldsymbol{p} \in \mathcal{P}_0} \{\boldsymbol{p}^\top \boldsymbol{\ell}_\theta - \tau - \kappa_\tau(\theta)d(\boldsymbol{p}, \hat{\boldsymbol{p}})\} = 0, \tag{8}$$

with the supremum attained at $\boldsymbol{p}^* \in \mathcal{P}_0$. We write the Lagrangian as

$$\mathcal{L}(\boldsymbol{p}, \lambda) = \boldsymbol{p}^\top \boldsymbol{\ell}_\theta - \tau - \kappa_\tau(\theta)d(\boldsymbol{p}, \hat{\boldsymbol{p}}) + \lambda(\boldsymbol{1}^\top \boldsymbol{p} - 1), \tag{9}$$

with $\lambda \in \mathbb{R}$ for the normalization constraint. Throughout this section we assume that $d(\cdot, \cdot)$ is a Bregman divergence whose generator $\phi$ is Legendre-type with $\operatorname{int}(\operatorname{dom}\phi) \subseteq \mathbb{R}^n_{++}$ (Definition D.2 in Appendix D.2); the KL divergence

used in our experiments satisfies this assumption. Under this restriction we omit the nonnegativity constraints $p_i \geq 0$ from (9): as we show in Appendix D.2 that the nonnegativity constraints are inactive at the optimal solution.

The detailed step-by-step procedure is given in Algorithms 1 and 2. To improve computational efficiency, instead of considering all data when calculating $\kappa_\tau(\theta)$, we use a batch-based estimation. Let $\mathcal{B} \subset \mathcal{D}$ denote a mini-batch of size $|\mathcal{B}|$. In this case, $\hat{\boldsymbol{p}}$ is taken as the uniform distribution over the batch $\mathcal{B}$, and $\mathcal{P}_0$ is the probability simplex on the same support, that is,

$$\hat{\boldsymbol{p}} = \left(\tfrac{1}{|\mathcal{B}|}, \ldots, \tfrac{1}{|\mathcal{B}|}\right), \qquad \mathcal{P}_0 = \left\{\boldsymbol{p} \in \mathbb{R}_+^{|\mathcal{B}|} : \sum_{i=1}^{|\mathcal{B}|} p_i = 1\right\}.$$

This reduces the computational cost of evaluating $\kappa_\tau(\theta)$ to scale with the batch size $|\mathcal{B}|$ rather than the full dataset. As in standard mini-batch neural network training, the resulting update is a stochastic approximation to the full empirical fragility gradient.

**Definition 3.2** (Bregman Divergence). Let $\mathcal{C} \subseteq \mathbb{R}^m$ be a convex set, and let $\phi : \mathcal{C} \to \mathbb{R}$ be strictly convex and differentiable. In our application, $\mathcal{C}$ is typically the probability simplex $\mathcal{P}_0 \subseteq \mathbb{R}^n$ over the empirical support, or a lower-dimensional group simplex in the structured setting. The *Bregman divergence* generated by $\phi$ is the map $d_\phi : \mathcal{C} \times \mathcal{C} \to \mathbb{R}_+$ defined by

$$d_\phi(\boldsymbol{x}, \boldsymbol{y}) := \phi(\boldsymbol{x}) - \phi(\boldsymbol{y}) - \langle \nabla\phi(\boldsymbol{y}), \, \boldsymbol{x} - \boldsymbol{y} \rangle, \; \boldsymbol{x}, \boldsymbol{y} \in \mathcal{C}, \tag{10}$$

where $\nabla\phi(\boldsymbol{y})$ denotes the gradient of $\phi$ at $\boldsymbol{y}$.

The gradient of a Bregman divergence with respect to its first argument is

$$\nabla_{\boldsymbol{x}} d_\phi(\boldsymbol{x}, \boldsymbol{y}) = \nabla\phi(\boldsymbol{x}) - \nabla\phi(\boldsymbol{y}) . \tag{11}$$

Taking the derivative of the Lagrangian and equating to 0 we get

$$\nabla_{\boldsymbol{p}}\mathcal{L} = \boldsymbol{\ell}_\theta - \kappa_\tau(\theta)(\nabla\phi(\boldsymbol{p}) - \nabla\phi(\hat{\boldsymbol{p}})) + \lambda\mathbf{1} = \boldsymbol{0} \tag{12}$$

$$\iff \nabla\phi(\boldsymbol{p}) = \nabla\phi(\hat{\boldsymbol{p}}) + \frac{\boldsymbol{\ell}_\theta}{\kappa_\tau(\theta)} + \frac{\lambda}{\kappa_\tau(\theta)}\mathbf{1} . \tag{13}$$

Let $T(\boldsymbol{p}) = \nabla_{\boldsymbol{x}}\phi(\boldsymbol{x})|_{\boldsymbol{x}=\boldsymbol{p}}$. We show that $T(\cdot)$ is invertible for many Bregman divergences including the KL-divergence in the Appendix. Define the scalar-valued function $h(\boldsymbol{p}) = \frac{d(\boldsymbol{p},\hat{\boldsymbol{p}})}{\boldsymbol{p}^\top\boldsymbol{\ell}_\theta-\tau}$, such that $\frac{1}{h(\boldsymbol{p}^*)} = \kappa_\tau(\theta)$. We then write $\boldsymbol{p}^*$ as

$$T(\boldsymbol{p}^*) = T(\hat{\boldsymbol{p}}) + h(\boldsymbol{p}^*)\boldsymbol{\ell}_\theta + h(\boldsymbol{p}^*)\lambda\mathbf{1} \tag{14}$$

$$\implies \boldsymbol{p}^* = T^{-1}(T(\hat{\boldsymbol{p}}) + h(\boldsymbol{p}^*)\boldsymbol{\ell}_\theta + h(\boldsymbol{p}^*)\lambda\mathbf{1}). \tag{15}$$

Equation (15) shows that any stationary optimizer lies on a one-dimensional trajectory parameterized by the scalar

$h$. Instead of optimizing over the full vector $\boldsymbol{p} \in \mathbb{R}^n$, we introduce the function below, where $\lambda$ is chosen to satisfy $\mathbf{1}^\top c(h) = 1$: $c : \mathbb{R} \to \mathbb{R}^n$, $c(h) := T^{-1}(T(\hat{\boldsymbol{p}}) + h\boldsymbol{\ell}_\theta + h\lambda\mathbf{1})$, which parameterizes a curve in $\mathbb{R}^n$ along which the solution $\boldsymbol{p}^*$ must lie. The maximizer satisfies $c(h^*) = \boldsymbol{p}^*$ for $h^* = h(\boldsymbol{p}^*)$. Consequently, we can compute $\kappa_\tau(\theta)$ efficiently by maximizing $\frac{\boldsymbol{p}^\top\boldsymbol{\ell}_\theta-\tau}{d(\boldsymbol{p},\hat{\boldsymbol{p}})}$ over the one-dimensional trajectory defined by $c(h)$. The trajectory function $c(h)$ depends on the probability divergence used. Next lemma presents the trajectory for KL-divergence.

**Lemma 3.3.** *When $d(\cdot, \cdot)$ is the KL-divergence, the probability distribution $\boldsymbol{p}^*$ that achieves $\kappa_\tau(\theta)$ lies on the trajectory*

$$c_i(h) = \frac{\hat{p}_i e^{h\ell_{\theta,i}}}{\sum_{j=1}^n \hat{p}_j e^{h\ell_{\theta,j}}}, \quad h > 0 .$$

*Proof.* Subject to $\mathbf{1}^\top\boldsymbol{p} = 1$, consider the maximization

$$\sup_{\boldsymbol{p}\in\mathcal{P}_0} \left\{\boldsymbol{p}^\top\boldsymbol{\ell}_\theta - \kappa_\tau(\theta)\, D_{\mathrm{KL}}(\boldsymbol{p} \,\|\, \hat{\boldsymbol{p}})\right\},$$

The Lagrangian is

$$\mathcal{L}(\boldsymbol{p}, \lambda) = \sum_{i=1}^n \left[p_i\ell_{\theta,i} - \kappa_\tau(\theta)\left(p_i \log\tfrac{p_i}{\hat{p}_i} - p_i + \hat{p}_i\right)\right]$$
$$+ \lambda\left(\sum_{i=1}^n p_i - 1\right).$$

Stationarity with respect to $p_i$ gives

$$0 = \frac{\partial\mathcal{L}}{\partial p_i} = \ell_{\theta,i} - \kappa_\tau(\theta)\log\frac{p_i}{\hat{p}_i} + \lambda,$$

which rearranges to

$$p_i = \hat{p}_i \exp\left(\tfrac{\ell_{\theta,i}+\lambda}{\kappa_\tau(\theta)}\right).$$

Imposing the normalization constraint $\sum_i p_i = 1$ determines the constant $e^{\lambda/\kappa_\tau(\theta)}$, yielding

$$p_i^* = \frac{\hat{p}_i \exp(\ell_{\theta,i}/\kappa_\tau(\theta))}{\sum_{j=1}^n \hat{p}_j \exp(\ell_{\theta,j}/\kappa_\tau(\theta))}.$$

Setting $h = 1/\kappa_\tau(\theta)$ shows that $\boldsymbol{p}^*$ lies on the trajectory

$$c_i(h) = \frac{\hat{p}_i e^{h\ell_{\theta,i}}}{\sum_{j=1}^n \hat{p}_j e^{h\ell_{\theta,j}}}, \qquad h > 0,$$

as claimed. $\qquad\qquad\qquad\qquad\qquad\qquad\square$

Lemma 3.3 reduces the inner maximization over $\Delta^{n-1}$ to a one-dimensional search over $h > 0$ along the exponential-tilting path $c(h)$, so IRS(KL) computes $\boldsymbol{p}^* = c(h^*)$ via a scalar line search rather than optimizing over the full

---

**Algorithm 1** Iterative Robust Satisficing (IRS)

---

**Input:** dataset $\mathcal{D}$, loss $\ell(\theta, x, y)$, target threshold $\tau > 0$, learning rate $\eta > 0$, number of epochs $N_{\text{ep}}$, divergence $d_\phi(\cdot, \cdot)$ (Bregman), scheduling factor $\epsilon > 0$.

1: Initialize parameters $\theta_0$; set empirical distribution $\hat{\boldsymbol{p}} \leftarrow \widehat{\mathbb{P}}_{\mathcal{D}}$.
2: **for** $t = 0, 1, \ldots, N_{\text{ep}} - 1$ **do**
3:      Sample mini-batch $\mathcal{B} \subset \mathcal{D}$; form per-sample loss vector $\boldsymbol{\ell}_{\theta_t} \leftarrow \left[ \ell(\theta_t, x_i, y_i) \right]_{(x_i, y_i) \in \mathcal{B}}$.
4:      Compute in-distribution loss $L_{\widehat{\mathbb{P}}_{\mathcal{D}}}(\theta_t) \leftarrow \frac{1}{|\mathcal{B}|} \sum_{(x_i, y_i) \in \mathcal{B}} \ell(\theta_t, x_i, y_i)$.
5:      **(Threshold schedule)** $\tau_t \leftarrow \max\left\{ \tau, (1 + \epsilon) L_{\widehat{\mathbb{P}}_{\mathcal{D}}}(\theta_t) \right\}$.
6:      **(Inner maximization)** $\left( \boldsymbol{p}_t^*, \kappa_{\tau_t}(\theta_t) \right) \leftarrow \text{MAXIMIZEALONGCURVE}\left( \boldsymbol{\ell}_{\theta_t}, \hat{\boldsymbol{p}}, \tau_t, d_\phi \right)$.
7:      **if** $\kappa_{\tau_t}(\theta_t) > 0$ **then**
8:          **(Danskin / no backprop through argmax)** Compute gradient of fragility while treating $\boldsymbol{p}_t^*$ as a constant:

$$\nabla_\theta \kappa_{\tau_t}(\theta_t) = \frac{\nabla_\theta \left[ \boldsymbol{p}_t^{*\top} \boldsymbol{\ell}_{\theta_t} \right] \cdot d_\phi(\boldsymbol{p}_t^*, \hat{\boldsymbol{p}}) - \left( \boldsymbol{p}_t^{*\top} \boldsymbol{\ell}_{\theta_t} - \tau_t \right) \cdot \nabla_\theta d_\phi(\boldsymbol{p}_t^*, \hat{\boldsymbol{p}})}{d_\phi(\boldsymbol{p}_t^*, \hat{\boldsymbol{p}})^2} = \frac{\boldsymbol{p}_t^{*\top} \nabla_\theta \boldsymbol{\ell}_{\theta_t}}{d_\phi(\boldsymbol{p}_t^*, \hat{\boldsymbol{p}})}.$$

9:          **(Update)** $\theta_{t+1} \leftarrow \theta_t - \eta \nabla_\theta \kappa_{\tau_t}(\theta_t)$.
10:     **else**
11:         **(Skip)** $\theta_{t+1} \leftarrow \theta_t$.     *(mini-batch is already $\tau_t$-satisficing)*
12:     **end if**
13: **end for**
**Output:** Final parameters $\theta_{N_{\text{ep}}}$.

---

simplex. The full procedure is given in Algorithms 1 and 2. A second source of efficiency comes from the satisficing structure itself: whenever $\kappa_{\tau_t}(\theta_t) = 0$, the worst-case risk over $\mathcal{P}_0$ does not exceed the current threshold, the constraint is already met on the mini-batch, and the gradient step is skipped. Together, the scalar inner search and the skipped updates yield the wall-clock speedup reported in Section 4. We provide a convergence analysis of IRS under standard smoothness assumptions in Appendix D.4.

### 3.3. Adapting IRS to Structured Distribution Shifts

The IRS derivation in Sections 3.1 and 3.2 is presented in its most general, instance-wise form, where the reference distribution is the empirical distribution and the inner maximization ranges over the full probability simplex supported on the observed samples. Concretely, for a dataset (or a mini-batch) of size $n$, this corresponds to optimizing over $\Delta^{n-1}$, i.e., assigning a weight to each training instance. This formulation is fully nonparametric with respect to the shift type. It allows any reweighting of the observed support and therefore serves as a default choice when the structure of the deployment shift is unknown. In many benchmarks, however, we have domain knowledge that restricts the plausible shifts to a lower-dimensional family such as label shifts, group shifts, or shifts indexed by a context variable. IRS can exploit this structure by redefining the space over which the adversarial distribution ranges. Doing so can reduce the dimension of the inner problem and improve statistical stability, while retaining the same general framework.

**From instance-wise to group-wise IRS.** A convenient way to formalize structured shifts is to define a partition of the data into $G$ groups (classes, environments, or any user-defined context bins) and allow shifts only through the group mixture weights. Let $g(z) \in \{1, \ldots, G\}$ denote the group membership of an example $z = (x, y)$, and let $\hat{\boldsymbol{q}} \in \Delta^{G-1}$ be the reference group prior under the training distribution. For each group $j$, define the group-conditional expected loss $\bar{\ell}_j(\theta) := \mathbb{E}\left[ \ell(\theta, z) \mid g(z) = j \right]$. Under the restriction that distribution shift happens only by changing group proportions, the expected loss under a candidate deployment group prior $\boldsymbol{q} \in \Delta^{G-1}$ becomes $L_{\boldsymbol{q}}(\theta) = \boldsymbol{q}^\top \bar{\boldsymbol{\ell}}(\theta)$ where $\bar{\boldsymbol{\ell}}(\theta) = (\bar{\ell}_1(\theta), \ldots, \bar{\ell}_G(\theta))$. The corresponding *structured* fragility objective is then

$$\kappa_\tau^{(G)}(\theta) := \max_{\boldsymbol{q} \in \Delta^{G-1} \setminus \{\hat{\boldsymbol{q}}\}} \frac{\boldsymbol{q}^\top \bar{\boldsymbol{\ell}}(\theta) - \tau}{d(\boldsymbol{q}, \hat{\boldsymbol{q}})}. \tag{16}$$

This has the same form as the instance-wise objective in (4), but with the simplex dimension reduced from $n - 1$ to $G - 1$. When $G = n$ and each point forms its own group, Eq. (16) recovers the instance-wise IRS objective.

## 4. Experiments

### 4.1. Experimental Scope and Evaluation Principles

We evaluate IRS across three complementary settings designed to measure robustness under increasingly realistic forms of distribution shift. First, we use a controlled synthetic benchmark in which the magnitude of shift is explicitly parameterized, aimed for a proof-of-concept comparison

**Algorithm 2** MAXIMIZEALONGCURVE: 1D search for $\boldsymbol{p}^*$ and $\kappa$ on $c(h)$

---

**Input:** loss vector $\boldsymbol{\ell}_\theta \in \mathbb{R}^n$, empirical distribution $\hat{\boldsymbol{p}} \in \Delta^{n-1}$, threshold $\tau$, Bregman divergence $d_\phi$ with Legendre-type generator $\phi$, mapping $T = \nabla\phi$, and its inverse $T^{-1}$.

1: Define the curve $c(h)$ via the optimality condition (Lagrangian stationarity):

$$c(h,\lambda) := T^{-1}\big(T(\hat{\boldsymbol{p}}) + h\,\boldsymbol{\ell}_\theta + h\,\lambda\,\boldsymbol{1}\big).$$

2: For $h > 0$, choose $\lambda(h)$ such that $\boldsymbol{1}^\top c(h,\lambda(h)) = 1$. (normalization to simplex)

3: Define the 1D objective (fragility along the trajectory)

$$r(h) := \frac{c(h,\lambda(h))^\top \boldsymbol{\ell}_\theta - \tau}{d_\phi(c(h,\lambda(h)),\hat{\boldsymbol{p}})}.$$

4: **(1D maximization)** Find $h^* \in \arg\max_{h>0} r(h)$ using a scalar search (e.g., golden-section, grid+refine).

5: Set $\boldsymbol{p}^* \leftarrow c\big(h^*,\lambda(h^*)\big), \quad \kappa \leftarrow r(h^*)$.

**Output:** $\boldsymbol{p}^*$, $\kappa$.

---

of how test performance degrades as the test distribution departs from training. Second, we study long-tailed image classification on CIFAR-10-LT, where increasing class imbalance induces a label shift. Third, to address robustness under naturally occurring, domain-grounded shifts, we consider regression benchmarks where contextual and temporal variability produces feature and label shifts without synthetic perturbations. Our evaluation focuses on metrics commonly used to assess robustness under distribution shift. For long-tailed classification, we report overall accuracy, worst-class accuracy, and tail-class average accuracy, which directly reflect performance on minority classes and align naturally with our notion of fragility. For regression, we report test mean squared error (MSE). Within each benchmark, all methods share the same backbone and hyperparameter selection protocol. Our implementation is available at https://github.com/Bilkent-CYBORG/IRS

We compare IRS against standard ERM training, Sharpness-Aware Minimization (SAM) (Foret et al., 2021), Invariant Risk Minimization (IRM) (Arjovsky et al., 2019), DRO methods (GroupDRO (Sagawa et al., 2020), $\chi^2$-DRO and CVaR-DRO (Levy et al., 2020)), risk extrapolation methods (V-REx and MM-REx) (Krueger et al., 2021), and KL-divergence–based robust satisficing (KL-RS) (Yan et al., 2024). Detailed descriptions of the experimental protocols are provided in Appendix B. Additional experiments, including a ViT experiment, are reported in Appendix C.

**Choice of the satisficing level $\tau$.** The satisficing level $\tau$ controls the tolerance on in-distribution performance re-

quired before prioritizing robustness. Rather than tuning $\tau$ per benchmark, we fix it within broad problem classes. Specifically, we use $\tau = 0.1$ for image classification experiments (synthetic, CIFAR-10-LT, and Waterbirds), and $\tau = 0.5$ for tabular regression tasks. Importantly, $\tau$ is not tuned to maximize performance, but chosen conservatively to ensure stable training dynamics across datasets of the same type. A principled, data-dependent procedure for selecting $\tau$ remains an important direction for future work.

### 4.2. Synthetic Label-Shift Benchmark

We consider a controlled synthetic benchmark designed to isolate robustness under label-marginal shift. The data are generated from a Gaussian mixture model (GMM) with $K = 4$ classes in $\mathbb{R}^2$, where each class is an isotropic Gaussian with means placed on a circle of radius 2.0 and unit covariance. The training distribution consists of $N_{\text{train}} = 8000$ samples with mildly imbalanced class priors $[0.45, 0.25, 0.20, 0.10]$, forming the in-distribution data on which all models are trained. Distribution shift is induced by perturbing only the label marginals while keeping the class-conditional distributions $p(x \mid y)$ fixed. Specifically, we construct shifted class priors of the form

$$\boldsymbol{p}_s(y) \propto \hat{\boldsymbol{p}}(y)\exp\{s\,\boldsymbol{v}(y)\},$$

where $\hat{\boldsymbol{p}}$ denotes the training priors and $\boldsymbol{v} = [0, 0.25, 0.5, 0.75]$ defines the shift direction. For each target shift magnitude, measured by $\text{KL}(\boldsymbol{p}_s \,\|\, \hat{\boldsymbol{p}})$, we numerically solve for the corresponding tilt parameter $s$ and generate a test set of $N_{\text{test}} = 5000$ samples.

We compare ERM, SAM and IRS by training the same MLP architecture with two hidden layers of width 64 and ReLU activations, batch size 256, for 20 epochs. We repeat the experiment over 100 independent runs with different random seeds. Figure 1 reports the mean test accuracy with one standard deviation as the label shift increases. While ERM and SAM degrade rapidly under increasing shift, IRS maintains accuracy close to the in-distribution level, illustrating the benefit of direct fragility minimization in a controlled setting.

### 4.3. Long-Tailed Image Classification

We evaluate IRS on long-tailed image classification using CIFAR-10-LT, a standard benchmark for robustness under label-marginal shift. CIFAR-10-LT is constructed by imposing an exponential decay on class frequencies in the training set, while keeping the test set balanced.

**Dataset and setup.** Starting from CIFAR-10 (50,000 training and 10,000 test images across 10 classes), the long-tailed training distribution is generated with class counts

$$n_c = n_{\max} \cdot \text{IF}^{-c/(C-1)}, \quad c = 0, \ldots, C-1,$$

*Table 2.* CIFAR-10-LT test accuracy breakdown across imbalance factors (balanced test set). Tail = average accuracy over the three least frequent classes (horse, ship, truck); Worst = minimum per-class accuracy. ERM uses Adam for all imbalance factors.

| Method | IF = 1 (Balanced) | | | IF = 20 | | | IF = 50 | | | IF = 100 | | |
|---|---|---|---|---|---|---|---|---|---|---|---|---|
| | Avg | Tail | Worst | Avg | Tail | Worst | Avg | Tail | Worst | Avg | Tail | Worst |
| IRS | 0.885 | 0.928 | 0.763 | 0.702 | **0.582** | **0.553** | **0.632** | **0.433** | **0.388** | **0.575** | **0.285** | **0.174** |
| ERM | **0.890** | 0.934 | 0.759 | **0.707** | 0.555 | 0.470 | 0.622 | 0.355 | 0.239 | 0.550 | 0.208 | 0.088 |
| SAM | 0.740 | 0.797 | 0.536 | 0.663 | 0.565 | 0.303 | 0.421 | 0.081 | 0.000 | 0.378 | 0.024 | 0.000 |
| KL-RS | 0.786 | 0.853 | 0.620 | 0.406 | 0.060 | 0.000 | 0.343 | 0.000 | 0.000 | 0.340 | 0.011 | 0.000 |
| V-REx | 0.877 | 0.920 | 0.755 | 0.680 | 0.513 | 0.416 | 0.590 | 0.312 | 0.202 | 0.526 | 0.188 | 0.070 |
| MM-REx | **0.890** | 0.933 | 0.766 | 0.688 | 0.521 | 0.437 | 0.620 | 0.344 | 0.233 | 0.542 | 0.189 | 0.089 |
| IRM | 0.824 | 0.881 | 0.624 | 0.555 | 0.411 | 0.205 | 0.407 | 0.135 | 0.001 | 0.410 | 0.124 | 0.012 |
| GroupDRO | 0.886 | 0.919 | 0.771 | 0.462 | 0.161 | 0.042 | 0.608 | 0.359 | 0.282 | 0.519 | 0.193 | 0.123 |
| $\chi^2$-DRO | 0.831 | 0.876 | 0.620 | 0.654 | 0.522 | 0.289 | 0.604 | 0.409 | 0.273 | 0.480 | 0.260 | 0.125 |
| CVaR-DRO | **0.890** | **0.935** | **0.783** | 0.702 | 0.551 | 0.470 | 0.614 | 0.330 | 0.219 | 0.546 | 0.199 | 0.080 |

where $C = 10$, $n_{\max} = 5000$, and IF denotes the imbalance factor. The standard balanced CIFAR-10 test split (1,000 images per class) is used for evaluation.

All methods are trained using the same WideResNet-28-10 model and identical data preprocessing. A stratified subset of the long-tailed training data is held out for validation and hyperparameter selection. Full details of the training protocol, optimization settings, and numerical stabilization procedures are provided in the appendix.

**Discussion.** Table 2 highlights how robustness gaps widen as class imbalance increases. When the training distribution is balanced (IF = 1), the tail metric reduces to the average accuracy over a fixed subset of classes; robustness trends are therefore interpreted primarily from IF $\geq$ 20. For IF = 1, all methods achieve similar average accuracy, indicating that IRS does not sacrifice in-distribution performance. As imbalance grows (IF = 20, 50), however, clear differences emerge. While several baselines remain competitive in average accuracy, their performance deteriorates sharply on tail classes, as reflected in reduced tail averages and worst-class accuracy.

In contrast, IRS consistently achieves the strongest worst-class performance and among the highest tail averages, indicating a substantial reduction in tail-class fragility. Notably, KL-RS and SAM degrade severely under moderate to high imbalance. These trends become more pronounced under severe imbalance, and the same behavior persists at IF = 100, where IRS dominates across both average and imbalance-sensitive metrics. Overall, the results show that directly minimizing fragility yields models that remain reliable on minority classes under increasing label-marginal shift, outperforming sharpness-based, invariance-based, DRO-based, risk-extrapolation, and prior robust satisficing approaches, without incurring additional per-step computational cost.

*Table 3.* Waterbirds test performance under group shift. Worst-Group denotes accuracy on the worst-performing group.

| Method | Worst-Group (%) | Overall (%) |
|---|---|---|
| IRS (ours) | **88.01** | **91.77** |
| ERM | 63.08 | 86.85 |
| SAM | 65.42 | 87.80 |
| KL-RS | 73.83 | 89.21 |
| V-REx | 76.17 | 87.99 |
| MM-REx | 75.30 | 86.76 |
| IRM | 79.16 | 86.95 |
| GroupDRO | 85.51 | **92.13** |
| $\chi^2$-DRO | 73.68 | 85.59 |
| CVaR-DRO | 74.06 | 85.81 |

## 4.4. Natural and Group-Based Distribution Shifts

We next evaluate IRS under distribution shifts that arise naturally in real-world data and are not synthetically parameterized. These include group shifts induced by spurious correlations, as well as temporal and contextual shifts in tabular regression tasks.

### 4.4.1. WATERBIRDS

We evaluate IRS on the Waterbirds dataset, a standard benchmark for robustness under group shifts caused by spurious correlations between foreground objects and backgrounds. Performance is measured by worst-group accuracy, which captures failure on minority groups.

Table 3 reports test performance on Waterbirds under spurious correlation–induced group shift. While ERM and SAM achieve high overall accuracy, their worst-group performance is substantially lower, indicating sensitivity to background correlations. IRS markedly improves robustness, increasing worst-group accuracy from 63.08% (ERM) to

*Table 4.* Full-training wall-clock time on Waterbirds. All methods are fine-tuned for 50 epochs on a single 80GB A100 using an ImageNet-pretrained ResNet-50 and the same data pipeline. Speedup is computed relative to ERM (higher is better).

| Method | Wall-clock | Speedup vs. ERM |
|---|---|---|
| **IRS-Group** | **12:29.94** | **1.26×** |
| ERM | 15:45.32 | 1.00× |
| SAM | 20:09.77 | 0.78× |
| KL-RS | 17:57.05 | 0.88× |
| V-REx | 15:45.50 | 1.00× |
| MM-REx | 15:50.32 | 0.99× |
| IRMv1 | 16:00.00 | 0.98× |
| GroupDRO | 16:04.24 | 0.98× |
| $\chi^2$-DRO | 15:59.00 | 0.99× |
| CVaR-DRO | 15:59.11 | 0.99× |

*Table 5.* Tabular regression under natural distribution shifts. Test mean squared error (MSE). Mean and standard deviation over 10 independent runs are reported.

| Method | Bike Sharing | Concrete Strength |
|---|---|---|
| IRS (ours) | **0.815±0.020** | **1.116±0.020** |
| ERM | 1.009± 0.008 | 1.316±0.012 |
| SAM | 1.010±0.018 | 1.429±0.058 |
| KL-RS | 0.911 ± 0.048 | 1.235±0.461 |
| V-REx | 1.002±0.003 | 1.442±0.024 |
| MM-REx | 0.971±0.026 | 1.399±0.027 |
| IRM | 1.003±0.003 | 1.440±0.019 |
| GroupDRO | 0.991±0.007 | 1.297±0.016 |
| $\chi^2$-DRO | 0.860±0.015 | 1.309±0.010 |
| CVaR-DRO | 0.977±0.022 | 1.317±0.009 |

88.01%, while maintaining high overall accuracy (91.77%). Compared to GroupDRO, IRS attains higher worst-group accuracy (88.01% vs. 85.51%) with comparable overall accuracy (91.77% vs. 92.13%), demonstrating that fragility minimization can improve worst-group robustness without substantially sacrificing in-distribution performance.

**Wall-clock efficiency (Waterbirds).** Table 4 reports full-training wall-clock times on Waterbirds (50 epochs, ImageNet-pretrained ResNet-50, single 80GB A100; identical data pipeline across methods). IRS is faster than ERM and other robust baselines. This behavior is expected: IRS performs a parameter update only when the current mini-batch violates the satisficing threshold under $p^*$ (i.e., when the worst-case risk exceeds $\tau$); once a mini-batch is already $\tau$-satisficing over the considered distribution family, the update is skipped. Consequently, IRS typically executes fewer effective optimizer steps in practice, which translates into reduced wall-clock training time.

#### 4.4.2. TABULAR REGRESSION UNDER NATURAL DISTRIBUTION SHIFTS

We evaluate group-wise IRS on two tabular regression benchmarks that exhibit naturally occurring distribution shifts: Bike Sharing and Concrete Strength. In Bike Sharing, temporal and seasonal effects induce simultaneous feature and label shift; models are trained on three seasons and evaluated on the held-out season. In Concrete Strength, feature distributions vary with curing age, and models are trained on mid-range ages and evaluated on extreme ages. Performance is measured using test mean squared error (MSE). Table 5 shows that IRS consistently achieves the lowest test MSE on both datasets, outperforming ERM and robust baselines. These results demonstrate that the benefits of fragility minimization extend beyond vision and classification to regression problems with real-world, domain-driven distribution shifts.

## 5. Conclusion

We proposed *Iterative Robust Satisficing (IRS)*, a gradient-based method for minimizing fragility by directly optimizing a robust satisficing objective. The inner optimization admits a one-dimensional trajectory, yielding gradients at essentially the same per-step cost as SGD. Across multiple settings IRS consistently improves minority and worst-group performance while maintaining in-distribution accuracy, and can reduce wall-clock training time due to skipped updates once the satisficing threshold is met. The consistent gains on worst-group accuracy on Waterbirds and tail-class accuracy on CIFAR-10-LT, achieved without group annotations or additional forward-backward passes, suggest that fragility minimization captures a practically relevant notion of robustness that existing objectives miss.

These results support IRS as a practical alternative to ERM and DRO in settings where robustness is desired without excessive conservatism and the structure of distribution shift is unspecified. Beyond the considered experiments, IRS provides a general framework for trading off in-distribution performance and robustness through fragility minimization.

One limitation of the current formulation is that it is restricted to distribution shifts whose support is the training dataset. Moreover, when the target distribution is known, directly optimizing for that distribution may be more appropriate. Important directions for future work include principled selection of the satisficing level $\tau$, extensions of IRS to richer structured shifts with support beyond the training dataset and larger-scale training regimes, and combining IRS with data augmentation or representation learning objectives, both of which could extend the effective support of the adversarial distribution beyond the training set.

## Acknowledgements

This work was supported by the Scientific and Technological Research Council of Türkiye (TUBITAK) under Grant 124E065; TUBITAK 2024 Incentive Award; by the Turkish Academy of Sciences Distinguished Young Scientist Award Program (TUBA-GEBIP-2023).

## Impact Statement

This paper presents work whose goal is to advance the field of machine learning. There are many potential societal consequences of improving robustness to distribution shift, none of which we feel must be specifically highlighted here.

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

# A. Table of Notations

| Symbol | Description |
|---|---|
| $p$ | Number of input features. |
| $d$ | Number of learnable parameters. |
| $\theta \in \Theta \subseteq \mathbb{R}^d$ | Model Parameters. |
| $n$ | Number of training samples. |
| $\boldsymbol{x} \in \mathbb{R}^p$ | Input feature vector. |
| $\boldsymbol{X} \in \mathbb{R}^{p \times n}$ | Data matrix with $n$ samples. |
| $\mathcal{D} = \{z_i\}_{i=1}^n$ | Training dataset, with $z_i = (x_i, y_i)$. |
| $\mathbb{P}, \mathbb{Q}$ | Probability distributions on $\mathcal{X} \times \mathcal{Y}$. |
| $\mathbb{P}_d$ | Deployment distribution. |
| $\ell(\theta, x, y)$ | Per-sample loss (e.g., Mean Squared Error) for model parameter $\theta$ and sample $(x, y)$. |
| $\boldsymbol{\ell}_\theta$ | Per-sample loss vector $[\ell(\theta, z_1), \ldots, \ell(\theta, z_n)]^\top$. |
| $\widehat{\mathbb{P}}_\mathcal{D}$ | Reference distribution. |
| $\boldsymbol{p}$ | Probability vector over the empirical support. |
| $\hat{\boldsymbol{p}}$ | Probability-vector representation of the empirical/reference distribution. |
| $\Delta^{n-1}$ | Probability simplex over $n$ samples, $\{\boldsymbol{p} \in \mathbb{R}_+^n : \mathbf{1}^\top \boldsymbol{p} = 1\}$. |
| $\tau$ | Robust satisficing threshold. |
| $\mathcal{P}_0$ | Candidate distribution class; empirically, the probability simplex over samples. |
| $\mathbb{P}^*$ | The worst case distribution over the context space with a specific parameter $\theta$ for the robust satisficing objective, meaning the distribution that achieves $\kappa_\tau(\theta)$ |
| $d(\mathbb{P}_1, \mathbb{P}_2)$ | Divergence between two probability distributions $\mathbb{P}_1, \mathbb{P}_2 \in \mathcal{P}_0$. |
| $d_\phi(\boldsymbol{p}, \hat{\boldsymbol{p}})$ | Bregman divergence with generator function $\phi$ between probability vectors $(\boldsymbol{p}, \hat{\boldsymbol{p}})$. |
| $\phi$ | The generator function used to define a Bregman Divergence. |
| $T = \nabla\phi$ | Gradient function of the Bregman generator, $T(\boldsymbol{p}) = \nabla\phi(\boldsymbol{p})$. |
| $h$ | The scalar parameter of the one-dimensional trajectory $c(h)$. |
| $c(h)$ | One-dimensional trajectory used to search for $\boldsymbol{p}^*$. |
| $\epsilon$ | $\tau$-scheduling factor used at the beginning of training. |
| $\eta$ | Learning rate. |
| $\kappa_\tau(\theta)$ | Fragility of parameter $\theta$ at threshold $\tau$. |
| $G$ | Number of groups/classes in structured datasets. |
| $g(z)$ | Group assignment of example $z$. |
| $\boldsymbol{q}, \hat{\boldsymbol{q}}$ | Candidate and reference group-prior vectors. |
| $\bar{\ell}_j(\theta)$ | Expected loss for group $j$. |
| $\kappa_\tau^{(G)}(\theta)$ | Structured/group-wise fragility objective. |

*Table 6.* Notation used in this paper.

# B. Implementation Details

## B.1. Common Training Setup and Hyperparameter Selection

**Codebase and numerical precision.** All experiments are implemented in PyTorch. For vision benchmarks, we enable automatic mixed precision (AMP) for efficiency. For tabular regression we use full precision for numerical stability.

**Validation protocol.** Hyperparameters other than the aspiration level $\tau$ are selected via grid search on a held-out validation set specific to each benchmark. For methods with fixed recommended settings (e.g., SAM), we use the standard default values without additional tuning. We select the best configuration using the benchmark's primary validation metric: *worst-group accuracy* for Waterbirds, *overall accuracy* for CIFAR-10-LT, and *validation MSE* for regression.

**Choice of the aspiration level $\tau$.** In robust satisficing, $\tau$ specifies a *user-chosen performance floor* rather than a tunable hyperparameter. Selecting $\tau$ via i.i.d. validation is ill-posed: smaller $\tau$ always appears preferable under in-distribution validation and collapses the objective toward ERM. Accordingly, we fix $\tau$ *a priori* and do not tune it using validation or test data.

We choose $\tau$ in the natural scale of each loss with a consistent interpretation: (i) for **classification** (cross-entropy), we use $\tau = 0.1$, corresponding to an average true-label probability of $\exp(-0.1) \approx 0.90$ under a calibrated model; (ii) for **regression**, we standardize targets to unit variance so that MSE has a natural scale with a constant predictor achieving MSE $\approx 1$. We choose $\tau = 0.5$, corresponding to a nontrivial predictor that explains at least half of the target variance.

We use threshold scheduling only to stabilize early training when losses are large: at step $t$, we set $\tau_t = \max\{\tau, (1 + \epsilon)L_{\widehat{\mathbb{P}}_{\mathcal{D}}}(\theta_t)\}$ (cf. §3), and once $L_{\widehat{\mathbb{P}}_{\mathcal{D}}}(\theta_t) \leq \tau$ we continue with the fixed target $\tau$. Thus, $\epsilon$ affects only the initial stabilization phase; after the empirical loss falls below $\tau$, training proceeds with the fixed target threshold.

**IRS objective used in each benchmark (instance-/group-/class-wise).** IRS optimizes the RS fragility objective via an inner maximization over KL-divergence–constrained reweightings of a fixed reference distribution. Concretely, IRS takes one of the following forms depending on the shift structure:

- **Instance-wise IRS** (most general): the adversary reweights individual samples in the mini-batch (simplex over instances).

- **Group-wise IRS** (structured shifts): the adversary reweights a *small* number of groups $g \in \{1, \ldots, G\}$ and the loss is the corresponding mixture of group-conditional losses (simplex over groups), see Eq. (16).

- **Class-wise IRS** (label shift / long tails): a special case of group-wise IRS where groups are classes.

In our experiments we use **group-wise IRS** on Waterbirds (4 groups given in the dataset), **class-wise IRS** on CIFAR-10-LT (10 classes), and **group-wise IRS** on tabular regression using $G=5$ target-quantile bins.

**KL-RS.** KL-RS uses 1 epoch of ERM warmup to make $\tau$ feasible at the start of training. We evaluate KL-RS under the same benchmark training horizon used for the other methods. Because KL-RS performs an outer search over feasible fragility values, this horizon is split across its feasibility checks, whereas IRS follows a single gradient-based trajectory for the full horizon.

## B.2. Hyperparameter Grids

We report the full grids used for model selection in Table 7 (vision) and Table 8 (tabular regression). The chosen (best-validation) hyperparameters are reported in §B.3.

**Common learning-rate grid.** Unless otherwise stated, all methods search over lr $\in \{10^{-2}, 10^{-3}, 10^{-4}, 10^{-5}\}$ on the benchmark-specific validation set.

**SAM radius (vision).** Following the original SAM paper, we fix the perturbation radius to $\rho = 0.05$ for image classification, which was identified as a strong default via extensive empirical evaluation. Accordingly, we do not tune $\rho$ on validation or test data, and only tune the learning rate for SAM.

## B.3. Selected Hyperparameters

We report the best-validation hyperparameters selected from the grids in Table 7 and Table 8.

## B.4. Waterbirds (Group Shift)

**Benchmark and groups.** Waterbirds contains two labels (waterbird vs. landbird) and a spurious background attribute (water vs. land). We evaluate robustness using *worst-group accuracy* over the four groups defined by the pair $(y, b) \in \{0, 1\} \times \{0, 1\}$ (label $y$ and background $b$). We use the dataset-provided metadata and define a deterministic group id

$$g(y, b) = 2y + b \in \{0, 1, 2, 3\}.$$

*Table 7.* Hyperparameter grids used in vision benchmarks (Waterbirds, CIFAR-10-LT).

| Method | Grid |
|---|---|
| All methods | $\text{lr} \in \{10^{-2}, 10^{-3}, 10^{-4}, 10^{-5}\}$ |
| KL-RS | $\texttt{alt\_iters} \in \{1, 2, 3, 5, 10, 15\}$ |
| V-REx | $\beta \in \{0.1, 0.5, 1, 2, 10, 50\}$ |
| MM-REx | $\lambda \in \{0.1, 0.5, 1.0, 1.5, 2.0\}$ |
| IRMv1 | $\texttt{penalty\_weight} \in \{1, 10, 100\}$ |
| GroupDRO | $\alpha \in \{0.1, 0.2, 0.5\}$ |
| $\chi^2$-DRO | $\rho \in \{0.01, 0.1, 0.5, 1.0\}$ |
| CVaR-DRO | $\alpha \in \{0.05, 0.1, 0.2, 0.3, 0.5\}$ |

*Table 8.* Hyperparameter grids used in tabular regression benchmarks.

| Method | Grid |
|---|---|
| All methods | $\text{lr} \in \{10^{-2}, 10^{-3}, 10^{-4}, 10^{-5}\}$ |
| KL-RS | $\texttt{alt\_iters} \in \{1, 2, 3, 5, 10, 15\}$ |
| V-REx | $\beta \in \{0.01, 0.1, 1, 10, 50\}$ |
| MM-REx | $\lambda \in \{0.1, 0.5, 1.0, 2.0\}$ |
| IRMv1 | $\texttt{penalty\_weight} \in \{0.01, 0.1, 1, 10\}$ |
| GroupDRO | $\alpha \in \{0.1, 0.2, 0.5\}$ |
| $\chi^2$-DRO | $\rho \in \{0.01, 0.05, 0.1, 0.5, 1.0\}$ |
| CVaR-DRO | $\alpha \in \{0.05, 0.1, 0.2, 0.3, 0.5\}$ |

For methods requiring group/environment annotations (GroupDRO, IRM, and REx variants), we use this ground-truth group assignment.

**Model and optimization.** All methods fine-tune an ImageNet-pretrained ResNet-50 backbone for 50 epochs with batch size 256 using AMP. We use a shared weight decay ($10^{-4}$) and identical preprocessing across methods. Learning rates and method-specific parameters are selected from Table 7, and chosen values are reported in Table 9. The random seed 42 was used throughout the experiments. For all methods, Adam optimizer with the default momentum parameters $(\beta_1, \beta_2) = (0.9, 0.999)$ is used.

**IRS on Waterbirds.** We use **group-wise IRS** (Eq. (16)) with $G = 4$ groups. Let $\bar{\ell}_j(\theta)$ denote the empirical mean loss for group $j$ in the current mini-batch (computed using the group ids above), and let $\hat{q} \in \Delta^3$ be the training group prior computed once from the training split. IRS then solves the inner maximization over group priors

$$\max_{q \in \Delta^3 \setminus \{\hat{q}\}} \frac{q^\top \bar{\ell}(\theta) - \tau_t}{D_{\mathrm{KL}}(q \,\|\, \hat{q})}.$$

We warm-start the scalar search using the previous mini-batch solution. Updates are skipped whenever the constructed worst-case risk is already $\tau_t$-satisficing. In the outer gradient $\nabla_\theta \kappa_\tau(\theta) = \left(p^{*\top} \nabla_\theta \bar{\ell}(\theta)\right) / D_{\mathrm{KL}}(p^* \,\|\, \hat{p})$ we omit the $1/D_{\mathrm{KL}}$ prefactor. Since this factor is non-negative, the descent direction is preserved, and dropping it absorbs a state-dependent rescaling into the learning rate while avoiding the singularity that arises when $p^* \to \hat{p}$.

### B.5. CIFAR-10-LT (Long-Tailed Label Shift)

**Dataset.** CIFAR-10-LT is obtained from CIFAR-10 (50,000 train / 10,000 test) by subsampling the training set into a long-tailed class distribution with imbalance factor IF $\in \{1, 20, 50, 100\}$ using the standard exponential rule

$$n_c = n_{\max} \cdot \text{IF}^{-c/(C-1)}, \qquad c = 0, \ldots, C - 1,$$

with $C = 10$ and $n_{\max} = 5000$ (head class). This yields a tail class size of $5000/\text{IF}$ (e.g., 250 for IF=20, 100 for IF=50, 50 for IF=100). The test set remains the standard balanced CIFAR-10 test split (1,000 images per class).

**Data splits.** For each IF, we construct the long-tailed training subset and hold out a stratified validation subset from it for model selection (stratified by class to preserve the long-tailed proportions). Some methods require a sample from each

*Table 9.* Selected hyperparameters (Waterbirds).

| Method | Hyperparameters |
|---|---|
| ERM | lr $= 10^{-5}$ |
| SAM | lr $= 10^{-5}$ |
| IRS (group-wise) | lr $= 10^{-4}$ |
| KL-RS | lr $= 10^{-4}$, `alt_iters`$= 2$ |
| V-REx | lr $= 10^{-4}$, $\beta = 10$ |
| MM-REx | lr $= 10^{-4}$, $\lambda = 1$ |
| IRMv1 | lr $= 10^{-4}$, `penalty_weight`$= 100$ |
| GroupDRO | lr $= 10^{-4}$, $\alpha = 0.2$ |
| $\chi^2$-DRO | lr $= 10^{-4}$, $\rho = 1$ |
| CVaR-DRO | lr $= 10^{-4}$, $\alpha = 0.05$ |

*Table 10.* Selected learning rates for CIFAR-10-LT across imbalance factors (IF).

| Method | IF=1 | IF=20 | IF=50 | IF=100 |
|---|---|---|---|---|
| ERM | $10^{-3}$ | $10^{-3}$ | $10^{-3}$ | $10^{-3}$ |
| SAM | $10^{-3}$ | $10^{-3}$ | $10^{-3}$ | $10^{-3}$ |
| IRS (class-wise) | $10^{-3}$ | $10^{-3}$ | $10^{-3}$ | $10^{-3}$ |
| KL-RS | $10^{-3}$ | $10^{-3}$ | $10^{-3}$ | $10^{-4}$ |
| V-REx | $10^{-3}$ | $10^{-3}$ | $10^{-3}$ | $10^{-3}$ |
| MM-REx | $10^{-3}$ | $10^{-3}$ | $10^{-3}$ | $10^{-3}$ |
| IRMv1 | $10^{-4}$ | $10^{-4}$ | $10^{-4}$ | $10^{-4}$ |
| GroupDRO | $10^{-3}$ | $10^{-4}$ | $10^{-3}$ | $10^{-3}$ |
| $\chi^2$-DRO | $10^{-3}$ | $10^{-3}$ | $10^{-3}$ | $10^{-3}$ |
| CVaR-DRO | $10^{-3}$ | $10^{-3}$ | $10^{-3}$ | $10^{-3}$ |

class/group/environment in each batch. Hyperparameter grids are in Table 7; the selected values are reported in Table 10 and Table 11.

**Model and optimization.** All methods use WideResNet-28-10 and are trained for 100 epochs with batch size 512 using AMP. The fixed random seed 123 was used throughout the experiments. For SAM, we use the standard SGD base optimizer following the original SAM formulation, and tune its learning rate using the same grid as the other methods. For all other methods, Adam optimizer with the default momentum parameters $(\beta_1, \beta_2) = (0.9, 0.999)$ is used.

**Groups and pseudo-environments (for group-/environment-based baselines).** CIFAR-10-LT provides no environment metadata. We therefore use the following deterministic constructions:

- **Groups (for GroupDRO / IRS):** one group per class, i.e., $g(x, y) = y$ with $G = 10$.

- **Pseudo-environments (for IRM / REx variants):** CIFAR-10-LT provides no environment labels, so we construct $E = 2$ pseudo-environments via class-wise bootstrap resampling from the long-tailed training set (for the given imbalance factor). Let $C = 10$ and for each class $y \in \{1, \ldots, C\}$ let $\mathcal{I}_y$ be the index set of long-tailed training examples with label $y$, with $n_y = |\mathcal{I}_y|$. For each environment $e \in \{1, 2\}$ and each class $y$, we sample (with replacement) a multiset $\mathcal{S}_{e,y}$ of size $n_y$ from $\mathcal{I}_y$. We then form environment $e$ as the multiset concatenation $\mathcal{S}_e := \biguplus_{y=1}^{C} \mathcal{S}_{e,y}$.

**IRS on CIFAR-10-LT.** We use **class-wise IRS** (group-wise IRS with $G = 10$ classes). The reference prior $\hat{q}$ is the empirical training class prior induced by the long-tailed construction for the given IF

### B.6. Tabular Regression under Natural Shifts

**Benchmarks and metrics.** We evaluate robustness under naturally occurring distribution shifts in tabular regression using two UCI datasets: *Bike Sharing* and *Concrete Compressive Strength*. We report test mean squared error (MSE).

*Table 11.* Selected method-specific parameters for CIFAR-10-LT (continued from Table 10).

| Method | IF=1 | IF=20 | IF=50 | IF=100 |
|---|---|---|---|---|
| SAM | $\rho=0.05$ | $\rho=0.05$ | $\rho=0.05$ | $\rho=0.05$ |
| KL-RS | `alt`$=5$ | `alt`$=10$ | `alt`$=5$ | `alt`$=15$ |
| V-REx | $\beta=1.0$ | $\beta=1.0$ | $\beta=0.5$ | $\beta=0.5$ |
| MM-REx | $\lambda=2.0$ | $\lambda=1.0$ | $\lambda=1.5$ | $\lambda=1.5$ |
| IRMv1 | `pen`$=100$ | `pen`$=100$ | `pen`$=100$ | `pen`$=100$ |
| GroupDRO | $\alpha=0.1$ | $\alpha=0.5$ | $\alpha=0.1$ | $\alpha=0.2$ |
| $\chi^2$-DRO | $\rho=1.0$ | $\rho=0.1$ | $\rho=0.01$ | $\rho=0.5$ |
| CVaR-DRO | $\alpha=0.3$ | $\alpha=0.5$ | $\alpha=0.5$ | $\alpha=0.5$ |

*Table 12.* Selected hyperparameters for tabular regression.

| Method | Bike | Concrete |
|---|---|---|
| ERM | lr=1e-3 | lr=1e-4 |
| SAM | lr=1e-3, $\rho=0.05$ | lr=1e-4, $\rho=0.2$ |
| IRS (group-wise) | lr=1e-2 | lr=1e-4 |
| KL-RS | lr=1e-3, `alt_iters=1` | lr=1e-4, `alt_iters=10` |
| V-REx | lr=1e-3, $\beta=0.1$ | lr=1e-4, $\beta=0.1$ |
| MM-REx | lr=1e-2, $\lambda=1.0$ | lr=1e-4, $\lambda=1.0$ |
| IRMv1 | lr=1e-3, `penalty_weight=0.01` | lr=1e-4, `penalty_weight=0.1` |
| GroupDRO | lr=1e-3, $\alpha=0.1$ | lr=1e-4, $\alpha=0.5$ |
| $\chi^2$-DRO | lr=1e-2, $\rho=0.5$ | lr=1e-4, $\rho=0.1$ |
| CVaR-DRO | lr=1e-2, $\alpha=0.5$ | lr=1e-4, $\alpha=0.3$ |

**Preprocessing.** Inputs are standardized using `StandardScaler`. Targets are standardized to zero mean and unit variance.

**Environments (shift construction).** We construct two training environments and a shifted test environment:

- **Bike Sharing:** training uses `year=0` with environments defined by seasons $\{1, 2\}$ vs. $\{3, 4\}$, and testing uses `year=1`.

- **Concrete:** environments are defined by the `Age` feature: train on `Age` $< 28$ (Env-1) and `Age` $= 28$ (Env-2), and test on `Age` $> 28$.

Within each training environment, we randomly split data into train/validation (80/20) for model selection.

**Pseudo-groups for group-aware objectives.** To apply group-aware objectives (IRS, GroupDRO, and DRO baselines) in regression, we discretize the *training* targets into $G = 5$ bins via equal-frequency (quantile) binning. The bin edges are computed on the training portion only and then reused to assign group ids in validation/test to avoid leakage.

**Models and optimization.** For *Bike Sharing*, we use a degree-2 polynomial regressor (linear model on concatenated $x$ and $x^2$). For *Concrete*, we use a 3-layer MLP with two hidden layers of width 256 and ReLU activations. All methods train for 100 epochs with batch size 256. Method-specific parameters are selected by the grids in Table 8, with selected values in Table 12. The random seed 42 was used for hyperparameter search and seeds $42 - 52$ was used for 10 independent runs to give mean $\pm$ std results with the selected hyperparameters. For SAM, we use the standard SGD base optimizer following the original SAM formulation, and tune its learning rate using the same grid as the other methods. For all other methods, Adam optimizer with the default momentum parameters $(\beta_1, \beta_2) = (0.9, 0.999)$ is used.

**IRS and KL-RS configuration.** IRS is applied in a **group-wise** form over the $G = 5$ target-quantile groups, using the same KL-trajectory 1D inner maximization as in vision. KL-RS alternates inner/outer updates for the specified `alt_iters` (Table 12), and uses 3 ERM epochs as warmup, to make $\tau$ feasible for the model.

# C. Additional Experiments

## C.1. DomainBed TerraIncognita

We also ran an additional domain-generalization experiment on DomainBed TerraIncognita, a camera-trap image dataset with four location domains and 10 classes (Gulrajani & Lopez-Paz, 2021). We use location 100 as the held-out test domain and train on the remaining locations, reserving 10% of the training examples for validation. Location labels are used only to define the split and are not provided to the training objectives, so we compare ERM, SAM, CVaR-DRO, and instance-wise IRS. All methods fine-tune an ImageNet-1k-pretrained DeiT-S/16 backbone (Touvron et al., 2021) for 10 epochs using Adam optimizer with the default momentum parameters $(\beta_1, \beta_2) = (0.9, 0.999)$, batch size 256, learning rate $2.5 \times 10^{-4}$, and no weight decay. We report final-epoch test accuracy on the held-out location in Table 13.

For all other methods, Adam optimizer with the default momentum parameters $(\beta_1, \beta_2) = (0.9, 0.999)$ is used.

*Table 13.* Additional DomainBed TerraIncognita held-out-location test accuracy (%).

| Method | Test Accuracy (%) |
|---|---|
| IRS (ours) | **51.81** |
| SAM | 50.97 |
| ERM | 47.92 |
| CVaR-DRO | 21.34 |

## C.2. Learning curves under label shift: delayed generalization gains of IRS

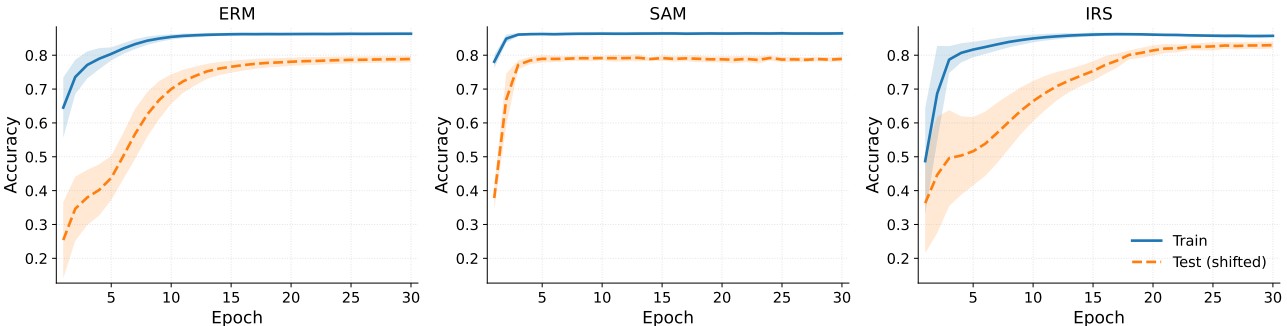

*Figure 2.* **Training vs. shifted-test learning curves under label shift.** Accuracy is shown as a function of epoch under a fixed label shift with $\mathrm{KL}(p_{\text{test}}(y) \| p_{\text{train}}(y)) = 0.80$, averaged over $N = 10$ runs (shaded regions: $\pm 1$ std). While ERM and SAM largely plateau on the shifted test set once training accuracy stabilizes, IRS continues to improve in shifted-test accuracy even after training accuracy has converged, indicating a form of implicit regularization.

Figure 2 reports training and shifted-test accuracy as a function of epoch under a fixed label shift with $\mathrm{KL}(p_{\text{test}}(y) \| p_{\text{train}}(y)) = 0.80$, averaged over $N = 10$ random runs (shaded regions show $\pm 1$ standard deviation).

Across ERM and SAM, training accuracy increases rapidly and then saturates; however, the shifted-test accuracy largely plateaus once training performance stabilizes. In contrast, IRS exhibits a qualitatively different behavior: even after the training accuracy has effectively converged, the shifted-test accuracy continues to improve over subsequent epochs. This separation between (i) stabilization of training accuracy and (ii) continued gains on the shifted test distribution is consistent with a form of *implicit regularization*: IRS keeps adjusting parameters in directions that have negligible effect on training classification accuracy but systematically improve robustness to the label-prior shift. Put differently, once the classifier is already fitting the training labels, further progress for IRS appears to come from reducing sensitivity to spurious reliance on the training label marginals, yielding better out-of-distribution performance without sacrificing training accuracy.

# D. Proofs

### D.1. Ratio form of fragility

We derive (4) from the constrained definition

$$\kappa_\tau(\theta) \;=\; \min\Big\{k \geq 0 \;:\; L_\mathbb{P}(\theta) \leq \tau + k\, d(\mathbb{P}, \widehat{\mathbb{P}}_\mathcal{D}) \;\; \forall \mathbb{P} \in \mathcal{P}_0\Big\}. \tag{17}$$

Throughout, $\mathcal{P}_0$ is the simplex over the atoms $\{z_i\}_{i=1}^n$, $L_\mathbb{P}(\theta) = \boldsymbol{p}^\top \boldsymbol{\ell}_\theta$ with $\boldsymbol{\ell}_\theta = (\ell(\theta, z_i))_{i=1}^n$, and $d$ is a Bregman divergence with $d(\mathbb{P}, \widehat{\mathbb{P}}_\mathcal{D}) = 0$ iff $\mathbb{P} = \widehat{\mathbb{P}}_\mathcal{D}$.

**Lemma D.1** (Three regimes of fragility). *Under the standing assumptions of Section 3,*

*(i) If $L_{\widehat{\mathbb{P}}_\mathcal{D}}(\theta) > \tau$, then $\kappa_\tau(\theta) = \infty$.*

*(ii) If $\max_i \ell(\theta, z_i) \leq \tau$, then $\kappa_\tau(\theta) = 0$.*

*(iii) If $L_{\widehat{\mathbb{P}}_\mathcal{D}}(\theta) < \tau$ and $\max_i \ell(\theta, z_i) > \tau$, then*

$$\kappa_\tau(\theta) \;=\; \max_{\mathbb{P} \in \mathcal{P}_0 \setminus \{\widehat{\mathbb{P}}_\mathcal{D}\}} \frac{L_\mathbb{P}(\theta) - \tau}{d(\mathbb{P}, \widehat{\mathbb{P}}_\mathcal{D})}, \tag{18}$$

*and the maximum is attained.*

*(iv) If $L_{\widehat{\mathbb{P}}_\mathcal{D}}(\theta) = \tau$ and $\boldsymbol{\ell}_\theta \not\propto \mathbf{1}$, then $\kappa_\tau(\theta) = \infty$.*

*Proof.* **(i).** At $\mathbb{P} = \widehat{\mathbb{P}}_\mathcal{D}$ the constraint reduces to $L_{\widehat{\mathbb{P}}_\mathcal{D}}(\theta) \leq \tau$, violated by assumption; no finite $k$ works.

**(ii).** For every $\boldsymbol{p} \in \mathcal{P}_0$, $L_\mathbb{P}(\theta) = \sum_i p_i \ell(\theta, z_i) \leq \max_j \ell(\theta, z_j) \leq \tau$, so the constraint holds at $k = 0$.

**(iii).** The constraint at $\widehat{\mathbb{P}}_\mathcal{D}$ holds for every $k \geq 0$. For $\mathbb{P} \neq \widehat{\mathbb{P}}_\mathcal{D}$, $d(\mathbb{P}, \widehat{\mathbb{P}}_\mathcal{D}) > 0$ and the constraint is equivalent to $k \geq (L_\mathbb{P}(\theta) - \tau)/d(\mathbb{P}, \widehat{\mathbb{P}}_\mathcal{D})$, giving

$$\kappa_\tau(\theta) \;=\; \max\Big\{0, \;\; \sup_{\mathbb{P} \in \mathcal{P}_0 \setminus \{\widehat{\mathbb{P}}_\mathcal{D}\}} \frac{L_\mathbb{P}(\theta) - \tau}{d(\mathbb{P}, \widehat{\mathbb{P}}_\mathcal{D})}\Big\}. \tag{19}$$

Let $i^\star \in \arg\max_i \ell(\theta, z_i)$ and $\mathbb{P}^\star = \delta_{z_{i^\star}}$. Then $L_{\mathbb{P}^\star}(\theta) - \tau > 0$ and $d(\mathbb{P}^\star, \widehat{\mathbb{P}}_\mathcal{D}) > 0$, so the supremum is strictly positive and the outer max with 0 is redundant.

For attainment, Assumption A2 gives a unique maximizer $\mathbb{P}^*(\theta)$ with $d(\mathbb{P}^*(\theta), \widehat{\mathbb{P}}_\mathcal{D}) > 0$. Setting $\epsilon = \frac{1}{2} d(\mathbb{P}^*(\theta), \widehat{\mathbb{P}}_\mathcal{D})$, the set $\mathcal{P}_\epsilon = \{\mathbb{P} \in \mathcal{P}_0 : d(\mathbb{P}, \widehat{\mathbb{P}}_\mathcal{D}) \geq \epsilon\}$ is compact, the ratio is continuous on it, and the maximizers over $\mathcal{P}_\epsilon$ and $\mathcal{P}_0 \setminus \{\widehat{\mathbb{P}}_\mathcal{D}\}$ coincide.

**(iv).** Equation (19) still applies. Pick $\boldsymbol{v} \in \mathbb{R}^n$ with $\mathbf{1}^\top \boldsymbol{v} = 0$ and $\boldsymbol{v}^\top \boldsymbol{\ell}_\theta > 0$ (possible since $\boldsymbol{\ell}_\theta \not\propto \mathbf{1}$). For small $a > 0$, $\boldsymbol{p}_a = \hat{\boldsymbol{p}} + a\boldsymbol{v} \in \mathcal{P}_0$ and

$$L_{\mathbb{P}_a}(\theta) - \tau \;=\; a\, \boldsymbol{v}^\top \boldsymbol{\ell}_\theta \;>\; 0, \qquad d(\mathbb{P}_a, \widehat{\mathbb{P}}_\mathcal{D}) \;=\; \tfrac{1}{2}a^2\, \boldsymbol{v}^\top \nabla^2 \phi(\hat{\boldsymbol{p}})\boldsymbol{v} + O(a^3)$$

(Taylor expansion of $d$; the leading coefficient is positive by strict convexity of $\phi$). Hence the ratio is $\Theta(1/a) \to \infty$ and the supremum in (19) is $+\infty$. $\square$

Regime (iii) is (4). Regimes (i) and (iv) justify the $\kappa_\tau = \infty$ convention on $L_{\widehat{\mathbb{P}}_\mathcal{D}}(\theta) \geq \tau$; regime (ii) justifies the skip rule in Algorithm 1.

## D.2. Invertibility of $T(p)$, positivity and optimality of $p^*$

**Definition D.2** (Legendre-type generator on the nonnegative orthant). Let $\phi : \mathbb{R}^n \to (-\infty, \infty]$ be proper, closed, and convex with $\partial(\mathrm{dom}\,\phi) \supseteq \mathrm{relbd}(\mathcal{P}_0)$ and $\mathrm{int}(\mathrm{dom}\,\phi) \supseteq \mathrm{relint}(\mathcal{P}_0)$. We say $\phi$ is *Legendre-type on the nonnegative orthant* if

1. **Essentially smooth:** $\phi$ is differentiable on $\mathrm{int}(\mathrm{dom}\,\phi)$ and for any sequence $x_k \to \partial(\mathrm{dom}\,\phi)$ we have $\|\nabla\phi(x_k)\| \to \infty$.

2. **Strictly convex:** $\phi$ is strictly convex on $\mathrm{int}(\mathrm{dom}\,\phi)$.

3. **Domain in the nonnegative orthant:** $\mathrm{int}(\mathrm{dom}\,\phi) \subseteq \mathbb{R}^n_{++}$, where $\mathbb{R}^n_{++} = \{x \in \mathbb{R}^n : x_i > 0 \text{ for all } i\}$ is the open positive orthant.

**Invertibility of $T(p)$**    Recall $T(p) = \nabla\phi(p)$. By Rockafellar (1970, Thm. 26.5), the essentially smooth and strictly convex properties of a Legendre-type generator imply that $T$ is a bijection from $\mathrm{int}(\mathrm{dom}\,\phi)$ onto $\mathrm{int}(\mathrm{dom}\,\phi^*)$. Since $\mathrm{int}(\mathrm{dom}\,\phi) \subseteq \mathbb{R}^n_{++}$, $T^{-1}(u)$ has strictly positive components for every admissible $u$.

**Positivity of $p^*$**    Define the objective function as $F(p) := p^\top \ell_\theta - \tau - \kappa_\tau(\theta)d(p, \hat{p})$. By definition of $\kappa_\tau(\theta)$, under the regime in Lemma D.1(iii), $\sup_{p \in \mathcal{P}_0} F(p) = 0$, and the supremum is attained at a distribution $p^* \neq \hat{p}$. Since $\kappa_\tau(\theta) > 0$, $d(p^*, \hat{p}) < \infty$. Hence, $p^* \in \mathrm{dom}\,\phi$.

We will prove that $p^*$ cannot lie on the relative boundary of the simplex. Suppose, for contradiction, that $p^* \in \mathrm{relbd}(\mathcal{P}_0)$ so it is also in $\partial(\mathrm{dom}\,\phi)$, thus, $p^* \in \partial(\mathrm{dom}\,\phi) \cap \mathrm{dom}\,\phi$ (for generators whose domain do not intersect with $\mathrm{relbd}(\mathcal{P}_0)$, this cannot happen, hence $p^*$ should be positive). Then there exists at least one coordinate $i$ such that $p_i^* = 0$. Fix $\varepsilon \in (0, 1]$. Since $\hat{p}_i > 0$ for all $i$, the point

$$p_\varepsilon = (1 - \varepsilon)p^* + \varepsilon\hat{p}$$

belongs to $\mathrm{relint}(\mathcal{P}_0)$ and $\mathrm{int}(\mathrm{dom}\,\phi)$.

Because $p^*$ maximizes $F$, we must have $F(p_\varepsilon) - F(p^*) \leq 0$. Let $h = \hat{p} - p^*$. Then $p_\varepsilon = p^* + \varepsilon h$. Using the definition of the Bregman divergence,

$$d(p, \hat{p}) = \phi(p) - \phi(\hat{p}) - \nabla\phi(\hat{p})^\top(p - \hat{p}),$$

we obtain

$$d(p_\varepsilon, \hat{p}) - d(p^*, \hat{p}) = \phi(p^* + \varepsilon h) - \phi(p^*) - \varepsilon\nabla\phi(\hat{p})^\top h.$$

Therefore,

$$\frac{F(p_\varepsilon) - F(p^*)}{\varepsilon} = h^\top\ell_\theta - \kappa_\tau(\theta)\left[\frac{\phi(p^* + \varepsilon h) - \phi(p^*)}{\varepsilon} - \nabla\phi(\hat{p})^\top h\right]. \tag{20}$$

Since $\phi$ is Legendre-type, it is essentially smooth and strictly convex. Therefore, at at any point in $\partial(\mathrm{dom}\,\phi) \cap \mathrm{dom}\,\phi$, the one-sided directional derivative toward the interior is $-\infty$. Therefore, since $p^* \in \partial(\mathrm{dom}\,\phi) \cap \mathrm{dom}\,\phi$, and $p_\varepsilon \in \mathrm{int}(\mathrm{dom}\,\phi)$, we have

$$\lim_{\varepsilon \downarrow 0} \frac{\phi(p^* + \varepsilon h) - \phi(p^*)}{\varepsilon} = -\infty.$$

Since $\kappa_\tau(\theta) > 0$, taking the limit of both sides of (20), we obtain

$$\lim_{\varepsilon \downarrow 0} \frac{F(p_\varepsilon) - F(p^*)}{\varepsilon} = +\infty.$$

Thus, for all sufficiently small $\varepsilon > 0$, $F(p_\varepsilon) > F(p^*)$, which contradicts the optimality of $p^*$. Hence, we proved that $p^* \in \mathrm{relint}(\mathcal{P}_0)$, or equivalently, $p_i^* > 0$ for all $i = 1, \ldots, n$.

The positivity of $p^*$ proven above implies that the nonnegativity constraints are inactive at $p^*$. Therefore, the first-order optimality condition at $p^*$ can be written using only the normalization constraint, justifying the focus on

$$\mathcal{L}(p, \lambda) = p^\top\ell_\theta - \tau - \kappa_\tau(\theta)d(p, \hat{p}) + \lambda(\mathbf{1}^\top p - 1).$$

Let $\tilde{\boldsymbol{p}}$ be the solution of the Lagrangian stationarity condition $\tilde{\boldsymbol{p}} = T^{-1}\big(T(\hat{\boldsymbol{p}}) + h(\tilde{\boldsymbol{p}})\boldsymbol{\ell}_\theta + h(\tilde{\boldsymbol{p}})\lambda\mathbf{1}\big)$ of Section 3.2 with $\lambda$ chosen such that $\mathbf{1}^\top\tilde{\boldsymbol{p}} = 1$. The above discussion with $\boldsymbol{u} = T(\hat{\boldsymbol{p}}) + h(\tilde{\boldsymbol{p}})\boldsymbol{\ell}_\theta + h(\tilde{\boldsymbol{p}})\lambda\mathbf{1}$ and $\tilde{\boldsymbol{p}} = T^{-1}(\boldsymbol{u})$ also verifies that $\tilde{\boldsymbol{p}} \in \mathbb{R}^n_{++}$.

**Optimality of $\boldsymbol{p}^*$.** Bregman divergence $d(\boldsymbol{p}, \hat{\boldsymbol{p}})$ with a Legendre-type generator $\phi$ is strictly convex in its first argument. Moreover, under the regime of interest (Lemma D.1(iii)) $\kappa_\tau(\theta) > 0$, hence the objective $F(\boldsymbol{p}) = \boldsymbol{p}^\top\boldsymbol{\ell}_\theta - \tau - \kappa_\tau(\theta)d(\boldsymbol{p}, \hat{\boldsymbol{p}})$ is strictly concave in $\boldsymbol{p}$. This implies that the feasible stationary point of the Lagrangian is the unique maximizer, hence $\boldsymbol{p}^* = \tilde{\boldsymbol{p}}$.

**KL example.** For the KL-divergence on the probability simplex the generator is $\phi(\boldsymbol{x}) = \sum_i x_i \log x_i$ with the convention $0 \log 0 = 0$, so $\operatorname{dom}\phi = \mathbb{R}^n_+$ and $\operatorname{int}(\operatorname{dom}\phi) = \mathbb{R}^n_{++}$. All three Legendre-type conditions are satisfied. Direct computation gives

$$T(\boldsymbol{x}) = (\log x_1 + 1, \ldots, \log x_n + 1), \qquad T^{-1}(\boldsymbol{u}) = \exp(\boldsymbol{u} - \mathbf{1}) \quad \text{(componentwise)},$$

both well-defined and componentwise positive, as required.

### D.3. Differentiating $\kappa_\tau(\theta)$ via Danskin's Theorem

**Theorem D.3** (Danskin's theorem, simple $C^1$ form). *Let $\Theta \subset \mathbb{R}^d$ be open, and let $\mathcal{P} \subset \mathbb{R}^n$ be nonempty, compact, and independent of $\theta$. Let $f : \Theta \times \mathcal{P} \to \mathbb{R}$ be continuous in $(\theta, \boldsymbol{p})$, and assume that for every fixed $\boldsymbol{p} \in \mathcal{P}$ the map $\theta \mapsto f(\theta, \boldsymbol{p})$ is $C^1$ with gradient $\nabla_\theta f(\theta, \boldsymbol{p})$. Define*

$$v(\theta) = \max_{\boldsymbol{p}\in\mathcal{P}} f(\theta, \boldsymbol{p}), \qquad S(\theta) = \arg\max_{\boldsymbol{p}\in\mathcal{P}} f(\theta, \boldsymbol{p}).$$

*If the maximizer is unique at $\theta$, i.e. $S(\theta) = \{\boldsymbol{p}^*(\theta)\}$, then $v$ is differentiable at $\theta$ and*

$$\nabla v(\theta) = \nabla_\theta f\big(\theta, \boldsymbol{p}^*(\theta)\big).$$

*In words: when differentiating the maximal value with respect to $\theta$, the maximizer can be treated as constant (no derivative through $\boldsymbol{p}^*(\theta)$). See Danskin (2012) and Bertsekas (2009, Proposition A.3.2) for proofs.*

*Remark* D.4 (Non-unique maximizers). When $S(\theta)$ is not a singleton, $v$ is in general only directionally differentiable and its Clarke subdifferential is the convex hull $\partial v(\theta) = \operatorname{co}\{\nabla_\theta f(\theta, \boldsymbol{p}) : \boldsymbol{p} \in S(\theta)\}$. Theorem D.3 then returns one element of $\partial v(\theta)$, which is sufficient for the gradient step in Algorithm 1.

**Application to the fragility gradient.** We apply Theorem D.3 to the fragility objective (Section 3). At any $\theta$ with $L_{\hat{\mathbb{P}}_\mathcal{D}}(\theta) < \tau$ and $\kappa_\tau(\theta) > 0$, and the fragility reduces to

$$\kappa_\tau(\theta) = \max_{\boldsymbol{p}\in\mathcal{P}_0\setminus\{\hat{\boldsymbol{p}}\}} f(\theta, \boldsymbol{p}), \qquad f(\theta, \boldsymbol{p}) := \frac{\boldsymbol{p}^\top\boldsymbol{\ell}_\theta - \tau}{d(\boldsymbol{p}, \hat{\boldsymbol{p}})}.$$

To apply Danskin's theorem cleanly we restrict the inner maximization to the compact, $\theta$-independent set

$$\mathcal{P}_\epsilon := \{\boldsymbol{p} \in \mathcal{P}_0 : d(\boldsymbol{p}, \hat{\boldsymbol{p}}) \geq \epsilon\}, \qquad \epsilon > 0.$$

Under Assumption A2 the maximizer $\boldsymbol{p}^*(\theta)$ lies in $\mathcal{P}_0 \setminus \{\hat{\boldsymbol{p}}\}$ and therefore satisfies $d(\boldsymbol{p}^*(\theta), \hat{\boldsymbol{p}}) > 0$. Pick any $\epsilon \in (0, d(\boldsymbol{p}^*(\theta), \hat{\boldsymbol{p}}))$; then $\boldsymbol{p}^*(\theta) \in \mathcal{P}_\epsilon$ and remains the unique maximizer of $f(\theta, \cdot)$ on $\mathcal{P}_\epsilon$, so

$$\kappa_\tau(\theta) = \max_{\boldsymbol{p}\in\mathcal{P}_\epsilon} f(\theta, \boldsymbol{p}).$$

On $\mathcal{P}_\epsilon$, $f$ is continuous in $(\theta, \boldsymbol{p})$, $C^1$ in $\theta$ for each fixed $\boldsymbol{p}$ (Assumption A1), the domain is compact and independent of $\theta$, and the maximizer is unique. Theorem D.3 therefore yields

$$\nabla\kappa_\tau(\theta) = \nabla_\theta f(\theta, \boldsymbol{p}^*)\Big|_{\boldsymbol{p}^*=\boldsymbol{p}^*(\theta)} = \frac{\big(\nabla_\theta\boldsymbol{\ell}_\theta\big)^\top\boldsymbol{p}^*\, d(\boldsymbol{p}^*, \hat{\boldsymbol{p}}) - \big(\boldsymbol{p}^{*\top}\boldsymbol{\ell}_\theta - \tau\big)\nabla_\theta d(\boldsymbol{p}^*, \hat{\boldsymbol{p}})}{d(\boldsymbol{p}^*, \hat{\boldsymbol{p}})^2}.$$

Because Danskin's theorem treats $\boldsymbol{p}^*$ as constant in $\theta$ and $\hat{\boldsymbol{p}}$ is the empirical distribution (also independent of $\theta$), the Bregman divergence $d(\boldsymbol{p}^*, \hat{\boldsymbol{p}})$ has no remaining $\theta$-dependence and $\nabla_\theta d(\boldsymbol{p}^*, \hat{\boldsymbol{p}}) = 0$. The cross term vanishes exactly, giving the simplified expression

$$\nabla \kappa_\tau(\theta) \;=\; \frac{\boldsymbol{p}^{*\top} \nabla_\theta \boldsymbol{\ell}_\theta}{d(\boldsymbol{p}^*, \hat{\boldsymbol{p}})}.$$

This is the formula used inside the gradient step of Algorithm 1, with $\tau$ replaced by the scheduled threshold $\tau_t$ that is treated as a stop-gradient scalar (see the discussion following Algorithm 1).

### D.4. Convergence of IRS

We next analyze gradient descent on a regularized fragility objective, where an additive $\delta > 0$ prevents vanishing denominators; in practice we also add a small $\delta$-stabilization to the KL term for numerical robustness.

**Theorem D.5** (Convergence of regularized IRS under smoothness). *Fix $\delta > 0$ and define the regularized fragility objective*

$$\tilde{\kappa}_\tau(\theta) \;:=\; \max_{\boldsymbol{p} \in \Delta^{n-1}} \frac{\boldsymbol{p}^\top \boldsymbol{\ell}_\theta - \tau}{D_{\mathrm{KL}}(\boldsymbol{p} \,\|\, \hat{\boldsymbol{p}}) + \delta}, \qquad \boldsymbol{\ell}_\theta = [\ell(\theta, z_i)]_{i=1}^n,$$

*where $\hat{\boldsymbol{p}}$ is fixed. Assume:*

(C1) *For each $i \in \{1, \ldots, n\}$, $\ell(\theta, z_i)$ is continuously differentiable in $\theta$.*

(C2) *For every $\theta$ considered, the maximizer set*

$$S(\theta) := \arg \max_{\boldsymbol{p} \in \Delta^{n-1}} \frac{\boldsymbol{p}^\top \boldsymbol{\ell}_\theta - \tau}{D_{\mathrm{KL}}(\boldsymbol{p} \| \hat{\boldsymbol{p}}) + \delta}$$

*is nonempty.*

(C3) *$\tilde{\kappa}_\tau$ is differentiable on the region visited by the iterates and has $L$-Lipschitz gradient there, i.e.,*

$$\|\nabla \tilde{\kappa}_\tau(\theta) - \nabla \tilde{\kappa}_\tau(\theta')\| \leq L \|\theta - \theta'\| \quad \text{for all } \theta, \theta' \text{ in that region.}$$

*Let $\{\theta_t\} \subset \mathbb{R}^d$ be generated by gradient descent*

$$\theta_{t+1} = \theta_t - \eta \nabla \tilde{\kappa}_\tau(\theta_t), \qquad 0 < \eta \leq \frac{1}{L}.$$

*Moreover, for each $t$, let $\boldsymbol{p}_t^* \in S(\theta_t)$ and compute $\nabla \tilde{\kappa}_\tau(\theta_t)$ using Danskin's theorem, i.e.*

$$\nabla \tilde{\kappa}_\tau(\theta_t) = \nabla_\theta \left. \frac{\boldsymbol{p}^\top \boldsymbol{\ell}_\theta - \tau}{D_{\mathrm{KL}}(\boldsymbol{p} \| \hat{\boldsymbol{p}}) + \delta} \right|_{\boldsymbol{p} = \boldsymbol{p}_t^*}.$$

*Then:*

1. *$\{\tilde{\kappa}_\tau(\theta_t)\}$ is non-increasing.*

2. *For all $T \geq 1$,*

$$\frac{1}{T} \sum_{t=0}^{T-1} \|\nabla \tilde{\kappa}_\tau(\theta_t)\|^2 \;\leq\; \frac{2\big(\tilde{\kappa}_\tau(\theta_0) - \inf_\theta \tilde{\kappa}_\tau(\theta)\big)}{\eta T}.$$

3. *Consequently, $\lim_{T \to \infty} \min_{0 \leq t < T} \|\nabla \tilde{\kappa}_\tau(\theta_t)\| = 0$.*

*Proof.* We first record a smoothness inequality implied by (C3). If a function $f$ has $L$-Lipschitz gradient on a convex set containing the line segment $\{\theta + s(\theta' - \theta) : s \in [0, 1]\}$, then

$$f(\theta') \leq f(\theta) + \langle \nabla f(\theta), \theta' - \theta \rangle + \frac{L}{2} \|\theta' - \theta\|^2. \tag{21}$$

A proof of this is included below for completeness.

Applying (21) to $f = \tilde{\kappa}_\tau$ with $\theta' = \theta - \eta \nabla \tilde{\kappa}_\tau(\theta)$ yields

$$\tilde{\kappa}_\tau(\theta - \eta \nabla \tilde{\kappa}_\tau(\theta)) \leq \tilde{\kappa}_\tau(\theta) - \eta \|\nabla \tilde{\kappa}_\tau(\theta)\|^2 + \frac{L}{2}\eta^2 \|\nabla \tilde{\kappa}_\tau(\theta)\|^2$$

$$= \tilde{\kappa}_\tau(\theta) - \eta\left(1 - \frac{L\eta}{2}\right)\|\nabla \tilde{\kappa}_\tau(\theta)\|^2.$$

With $0 < \eta \leq 1/L$, we have $1 - \frac{L\eta}{2} \geq \frac{1}{2}$, hence

$$\tilde{\kappa}_\tau(\theta - \eta \nabla \tilde{\kappa}_\tau(\theta)) \leq \tilde{\kappa}_\tau(\theta) - \frac{\eta}{2}\|\nabla \tilde{\kappa}_\tau(\theta)\|^2.$$

Setting $\theta = \theta_t$ gives monotonicity and, summing over $t = 0, \ldots, T-1$,

$$\frac{\eta}{2}\sum_{t=0}^{T-1} \|\nabla \tilde{\kappa}_\tau(\theta_t)\|^2 \leq \tilde{\kappa}_\tau(\theta_0) - \tilde{\kappa}_\tau(\theta_T) \leq \tilde{\kappa}_\tau(\theta_0) - \inf_\theta \tilde{\kappa}_\tau(\theta).$$

Dividing by $\eta T/2$ yields the stated average gradient-norm bound. Finally, $\min_{0 \leq t < T} \|\nabla \tilde{\kappa}_\tau(\theta_t)\|^2 \leq \frac{1}{T}\sum_{t=0}^{T-1} \|\nabla \tilde{\kappa}_\tau(\theta_t)\|^2$ implies the stationarity claim. $\square$

**Proof of** (21). Let $f$ have $L$-Lipschitz gradient. Define $\phi(s) = f(\theta + s(\theta' - \theta))$. Then $\phi'(s) = \langle \nabla f(\theta + s(\theta' - \theta)), \theta' - \theta \rangle$ and by Lipschitzness of $\nabla f$, $|\phi'(s) - \phi'(0)| \leq Ls\|\theta' - \theta\|^2$. Integrating from $s = 0$ to $1$ yields

$$f(\theta') - f(\theta) = \int_0^1 \phi'(s)\,ds \leq \phi'(0) + \int_0^1 Ls\|\theta' - \theta\|^2\,ds = \langle \nabla f(\theta), \theta' - \theta \rangle + \frac{L}{2}\|\theta' - \theta\|^2,$$

which is (21).

