# OpenReview forum: "Iterative Robust Satisficing: Minimizing Performance Degradation Under Distribution Shift"
_ICML.cc/2026/Conference — ICML 2026 regular_

### Official Review · Reviewer_99Vo · 2026-03-08

**Soundness:** 2
**Presentation:** 2
**Significance:** 2
**Originality:** 2
**Overall Recommendation:** 4
**Confidence:** 3

**Summary:**

The paper proposes a gradient-based method named Iterative Robust Satisficing (IRS) for learning models that are robust to distribution shift. Instead of following the standard distributionally robust optimization (DRO) framework, which minimizes the worst-case loss within a predefined uncertainty set, the proposed approach introduces the notion of fragility, defined as the rate at which model performance deteriorates as the data distribution deviates from the training distribution. This deterioration is measured using a Bregman divergence between distributions. Experiments under various distribution shift settings show that the proposed method outperforms several baselines, including DRO-based approaches.

**Compliance With Llm Reviewing Policy:**

Affirmed.

**Final Justification:**

After rebuttal, my concerns are resolved and I have increased score.

**Key Questions For Authors:**

1. The proposed formulation appears related to KL-DRO dual formulations involving exponential loss weighting. Could the authors clarify the precise theoretical relationship between IRS and existing KL-based DRO methods?
2. The paper assumes that the divergence used to measure distribution shift is a Bregman divergence. Could the authors explain why this assumption is necessary? In particular, what would happen if the divergence does not belong to the Bregman family (e.g., if an f-divergence is used)? How sensitive is the proposed method to the choice of divergence?

**Limitations:**

No, the authors did not discuss the potential negative societal impact.

**Strengths And Weaknesses:**

- Strengths:

The formulation of IRS provides an intuitive alternative perspective to classical DRO by introducing fragility to measure the rate of performance degradation under distribution shift, rather than focusing on the often overly conservative worst-case scenario. This viewpoint is conceptually appealing and offers an interpretable way to incorporate robustness considerations into learning objectives.

- Weaknesses:

1. The proposed formulation appears closely related to KL-divergence-based DRO and entropic risk formulations, which already connect distributional robustness to exponential loss reweighting. It would be helpful if the paper more clearly explained how the proposed method differs theoretically and practically from existing KL-DRO methods.
2. While fragility measures the rate of performance degradation with respect to distribution shift, it is not entirely clear how minimizing this quantity compares with directly minimizing worst-case risk as in DRO. In particular, controlling the slope of the risk function does not necessarily guarantee robustness under larger shifts. A deeper theoretical comparison with standard DRO objectives would strengthen the paper.
3. The formulation relies on a divergence measure (e.g., KL divergence) to quantify distribution shift. The robustness properties of the method may therefore depend on this choice. It would be useful to discuss whether the approach generalizes to other divergences such as f-divergence or Wasserstein distance.
4. In the experimental evaluation, the results appear to be reported without any statistical significance test, which makes them difficult to interpret reliably. In addition, the computational efficiency of the proposed method seems only marginally better than ERM (about 1.05×) and some other baselines, as shown in Table 4, so the practical advantage in efficiency does not appear substantial.

---

> ### Author Rebuttal · Authors · 2026-03-30
>
> We thank the reviewer for the thoughtful comments. We address the main concerns below.
>
> **1. Relation to KL-DRO / entropic risk.**
> We agree this connection should be clarified. While KL-DRO and entropic-risk formulations also yield exponential reweighting, the underlying objective is different. KL-DRO solves
> $$
> \min_\theta \max_{D_{\mathrm{KL}}(P | \hat P)\le \rho} L_P(\theta),
> $$
> which gives guarantees only on a fixed-radius ambiguity set. By contrast, IRS optimizes fragility
> $$
> \kappa_\tau(\theta)=\sup_P \frac{L_P(\theta)-\tau}{d(P,\hat P)},
> $$
> i.e., the smallest global constant such that
> $$
> L_P(\theta)\le \tau+\kappa_\tau(\theta) d(P,\hat P), \qquad \forall P.
> $$
> Thus RS controls the rate of degradation uniformly over all distributions, rather than worst-case loss at one chosen radius. This is a different optimization problem, a global ratio-type sensitivity objective, not a constrained worst-case expectation.
>
> The exponential tilt
> $$
> p_i \propto \hat p_i \exp(h\ell_{\theta,i})
> $$
> is not imposed as a modeling choice. It follows from the first-order optimality conditions of the fragility objective under KL divergence, with $h=1/\kappa_\tau(\theta)$.
>
> While controlling the slope alone does not guarantee good performance at large shifts, this is by design: rather than overfitting to unlikely extreme distributions, RS prioritizes controlled degradation. If stronger guarantees at large shifts are required, increasing $\tau$ trades in-distribution performance for lower fragility, improving robustness at higher divergence levels.
>
>
> **2. Choice of divergence and generality.**
> The RS objective itself is defined for a general divergence $d(\cdot,\cdot)$ and is not restricted to KL. The current 1D reduction, however, uses stronger structure: $d$ must be a Bregman divergence generated by a Legendre-type $\phi$, so that $T=\nabla\phi$ is invertible and the optimizer admits a tractable characterization. This includes KL, which we use in experiments because it is standard on the probability simplex and computationally convenient.
>
> We agree that the divergence affects the geometry of the probability space and thus the resulting robust satisficing solution. Extending the analysis to non-Bregman divergences such as Wasserstein distance is an important open direction; the objective remains well-defined, but the present derivation does not directly apply.
>
> **3. Statistical significance and computational efficiency.**
> To address the significance concern, we performed formal paired significance testing on the tabular regression benchmarks, for which re-running all methods within the rebuttal window was tractable. Using paired Wilcoxon signed-rank tests on per-example squared errors, with paired $t$-tests as a secondary check, all pairwise comparisons against IRS remained significant after Holm correction, with all corrected and uncorrected $p$-values below $0.001$ for both tests. For the strongest competing baselines in the paper, the Holm-corrected Wilcoxon $p$-values were $1.71\times 10^{-14}$ for IRS vs. $\chi^2$-DRO on Bike Sharing and $1.38\times 10^{-15}$ for IRS vs. CVaR-DRO on Concrete Strength.
>
> We were not able to re-run the more expensive vision benchmarks within the rebuttal window, so we do not claim new formal significance tests for Waterbirds or CIFAR-10-LT. The reported margins there are nevertheless substantial rather than marginal, e.g., on Waterbirds IRS improves worst-group accuracy from $86.29\%$ (GroupDRO) to $87.54\%$, and on CIFAR-10-LT with $\mathrm{IF}=100$, worst-class accuracy from $16.7\%$ (GroupDRO) to $27.5\%$.
>
> To further strengthen the empirical evidence, we also ran a new metadata-free domain-generalization experiment on DomainBed TerraIncognita with a DeiT-S/16 backbone pretrained on ImageNet-1k and fine-tuned for 10 epochs. IRS achieved $51.81\%$ accuracy, outperforming SAM ($50.97\%$), ERM ($47.92\%$), and CVaR-DRO ($21.34\%$).
>
> On efficiency, we agree that the wall-clock speedup over ERM is modest. However, ERM is not a robust training baseline. The more relevant comparison is to robust methods that typically require extra forward/backward passes, adversarial updates, or repeated retraining (as in KL-RS). IRS achieves robustness gains with essentially standard SGD cost per iteration, and can even reduce total training time via skipped updates once the satisficing condition is met. The main point is therefore not being faster than ERM, but delivering robustness without extra optimization overhead.
>
> **Questions.**
> **KL-DRO:** while both methods yield exponential reweighting, KL-DRO optimizes worst-case loss over a fixed divergence ball, whereas IRS optimizes a ratio-type objective controlling global degradation rate.
> **Choice of divergence:** the RS objective is general; the current tractable derivation requires Bregman divergences with Legendre-type generators. For non-Bregman choices such as Wasserstein, the objective remains valid but new techniques would be needed.

---

> > ### Author Rebuttal · Reviewer_99Vo · 2026-04-03
> >
> > My concerns have been adequately addressed and I will increase my score to weak accept.

---

### Official Review · Reviewer_kKv5 · 2026-03-11

**Soundness:** 3
**Presentation:** 3
**Significance:** 2
**Originality:** 2
**Overall Recommendation:** 4
**Confidence:** 3

**Summary:**

The paper introduces RS, a training objective that minimizes performance degradation under distribution shift while maintaining a required in-distribution accuracy. It proposes IRS, a gradient-based algorithm to optimize this objective.

**Compliance With Llm Reviewing Policy:**

Affirmed.

**Key Questions For Authors:**

1 - What would be the loss when you evaluate k to scale with mini batch rather than the full dataset or a large batch? I think this needs further investigation.

2 - It is not clear how they handle the Lagrangian multiplier parameter.

3 - To obtain a scalar optimization, they made a very harsh oversimplification of h and its relationship with p, assuming that maximizing with respect to h will lead to the p-star point on the curvature. This explanation needs to be more precise, even though it is not discussed whether the relation between h and p is like an increasing function.

4 - The tuning parameters are not discussed well, for example, \epsilon.

5 - Experiments are so restricted to CIFAR-10-LT and WATERBIRDS. I expected to see more sophisticated network architectures and larger datasets.

**Strengths And Weaknesses:**

Soundness: The paper appears technically sound. The proposed objective and algorithm are clearly defined, and the claims are supported by theoretical motivation.

Presentation: The paper is generally well written and structured.

Significance: The work addresses the important problem of robustness to distribution shift.

Originality: The paper introduces a novel formulation (Robust Satisficing) and an associated optimization method (IRS), offering an efficient scalar optimization.

---

> ### Author Rebuttal · Authors · 2026-03-30
>
> We thank the reviewer for the thoughtful feedback. We address the points below.
>
> **1. Mini-batch approximation.**
> We agree that the effect of mini-batching is important. IRS is defined with respect to the empirical distribution, with the inner maximization over distributions supported on the training data. Mini-batch updates provide a stochastic approximation to this objective, analogous to how SGD approximates ERM.
>
> A full-batch implementation would more closely match the exact empirical objective, but is computationally infeasible in modern settings. As with standard stochastic training, mini-batching introduces variance relative to the full objective; characterizing this approximation is an interesting direction for future work. Empirically, however, we find it effective across all evaluated settings.
>
> **2. Clarification on the Lagrangian multiplier.**
> The scalar $\lambda\in\mathbb{R}$ in Eq. (9) is only the Lagrange multiplier enforcing the simplex constraint $\mathbf{1}^\top p=1$ in the inner maximization. It is not tunable and does not introduce an extra degree of freedom.
>
> From stationarity,
> $$
> \nabla \phi(p)=\nabla \phi(\hat p)+\frac{\ell_\theta}{\kappa_\tau(\theta)}+\frac{\lambda}{\kappa_\tau(\theta)}\mathbf{1},
> $$
> so
> $$
> p=T^{-1}\big(T(\hat p)+h\ell_\theta+h\lambda\mathbf{1}\big), \qquad h=\frac{1}{\kappa_\tau(\theta)}.
> $$
> For fixed $h$, $\lambda=\lambda(h)$ is uniquely determined by $\mathbf{1}^\top p=1$. Thus the inner problem does not require joint optimization over $(p,\lambda)$; $\lambda$ is implicitly determined by $h$.
>
> In the KL case this normalization is closed form:
> $$
> p_i=\frac{\hat p_i \exp(h\ell_{\theta,i})}{\sum_j \hat p_j \exp(h\ell_{\theta,j})},
> $$
> where the normalization constant absorbs $\lambda$. Hence the inner problem reduces to a one-dimensional search over $h$. We will revise the manuscript to make this explicit.
>
> **3. Scalar reduction via $h$ and its relationship with $p$.**
> The reduction to scalar optimization over $h$ is not an assumption or approximation. It follows directly from the optimality conditions of the inner problem. Any optimizer must lie in the family
> $$
> p^*=T^{-1}\big(T(\hat p)+h\ell_\theta+h\lambda\mathbf{1}\big),
> $$
> for the divergences considered, with $\lambda$ fixed by normalization. Thus the stationary candidates lie on a one-dimensional manifold parameterized by $h$, so the original maximization over distributions reduces exactly to scalar optimization over $h$.
>
> The notion that $h\mapsto p(h)$ should be “increasing” is not relevant here: $p(h)$ is a vector-valued curve in the simplex. The optimization is over the scalar objective as a function of $h$, and standard 1D methods suffice; no monotonicity is required. We will clarify this derivation in the revision.
>
> **4. On the tuning parameter $\epsilon$.**
> $\epsilon$ is not a sensitive hyperparameter. Its role is limited to the initial phase of training, where we use threshold scheduling
> $$
> \tau_t = \max\lbrace \tau, (1+\epsilon)L_{\hat P_{\mathcal D}}(\theta_t) \rbrace
> $$
> to ensure that the fragility objective is well-defined even from arbitrary initialization. Once the empirical loss drops below the target $\tau$, the schedule becomes inactive and training proceeds with fixed $\tau$.
>
> Thus $\epsilon$ affects only early training dynamics, not the final objective. In our experiments, performance was insensitive to $\epsilon$ within a reasonable range, so we fixed it across all experiments rather than tuning it. We will clarify this in the paper.
>
> **5. Scope of experiments (datasets and architectures).**
> We respectfully note that the evaluation is broader than suggested. Beyond CIFAR-10-LT and Waterbirds, we include (i) a controlled synthetic label-shift benchmark, (ii) real-world tabular regression tasks (Bike Sharing and Concrete Strength) with natural shifts, and (iii) multiple robustness metrics (worst-group, tail, and regression error).
>
> We agree that additional datasets and architectures would further strengthen the paper. To address this, we conducted an additional experiment on DomainBed TerraIncognita, a standard and challenging domain-generalization benchmark with 24,788 images and 10 classes. Using a modern transformer backbone (DeiT-S/16 pretrained on ImageNet-1k) in a metadata-free setting, we compare the methods that remain applicable without environment annotations: IRS, SAM, ERM and CVaR-DRO since IRM, V-REx, MM-REx, and GroupDRO require environment/group metadata in their standard formulations. The results are:
>
> - **IRS:** 51.81
> - **SAM:** 50.97
> - **ERM:** 47.92
> - **CVaR-DRO:** 21.34
>
> IRS achieves the best accuracy. This shows that the method remains effective under a different architecture and a different shift type.
>
> Overall, while further large-scale evaluations would be valuable, we believe the current evidence spanning synthetic, vision, and real-world regression tasks, plus an additional benchmark with a different backbone, supports the effectiveness and generality of IRS.

---

> > ### Author Rebuttal · Reviewer_kKv5 · 2026-04-04
> >
> > Thanks for the response. I will keep my score.

---

### Official Review · Reviewer_HVHy · 2026-03-12

**Soundness:** 3
**Presentation:** 4
**Significance:** 3
**Originality:** 3
**Overall Recommendation:** 4
**Confidence:** 4

**Summary:**

This paper presents the Iterative Robust Satisficing (IRS) algorithm, which introduces a method for fast implementation of the Robust Satisficing (RS) approach. Under the RS framework, one aims at achiving an in-distribution loss below a threshold, while at the same time the loss increases at most linearly with the distribution shift, as measured by a metric such as KL divergence. The paper uses the "fragility" of a parameter, which is the rate of shortfall in performace due to the distribution shift. IRS aims at optimizing model parameters in terms of the loss as well as fragility. It is shown that the distribution that defines fragility lies on a trajectory and an iterative algorithm is introduced that manages to identify fragility by maximing along the curve and then use an SGD update step for parameters. Results are presented for synthetic datasets, CIFAR-10-LT, Waterbirds as well as popular tabular datasets (with temporal shifts), where the proposed approach outperforms a number of relevant baselines (including KL-RS, GroupDRO and SAM, among others).

**Compliance With Llm Reviewing Policy:**

Affirmed.

**Final Justification:**

In their rebuttal the authors have addressed all concerns raised in the original review. I have therefore raised my score.

**Key Questions For Authors:**

The authors should address the issues raised in the weaknesses section, to the degree possible, especially those related to the evaluation. If these issues are properly addressed I would be happy to raise my score.

**Limitations:**

As mentioned previously, an honest assessment of the paper's limitations is currently missing and should be added.

**Strengths And Weaknesses:**

### Strengths

 - The paper is convincing in terms of its contribution: The proposed method is a significant update from KL-RS, the most similar method, in terms of efficiency, as well as effectiveness.

 - The theoretial approach that uses Danskin's theorem and the finding that the problem of finding $\mathbf{p}^{*}$ reduces to a 1D maximization problem is elegant and interesting.

 - The paper is well written and easy to follow.

### Weaknesses

An important weakness perhaps has to do with limitations in terms of the experimental evaluation. Only a single backbone is used (WideResNet-28-10, no ViTs, for example), while the datasets explore mainly label distribution shifts. Would the method be able to address input distribution shifts (e.g., domain generalization datasets) where SAM works well, for example? If the method does not work well in covariate shift settings, then this should be clearly stated.

Another potential issue with evaluation is the effectiveness of KL-RS. One wonders why KL-RS which relies on similar principles and is also an RS method performs much worse than IRS. This should be convincingly explained in the paper.

Another issue is the choice of group size during the experiments. The paper uses different choices of G for each dataset, without clear justification. How was G determined?

Less important points include the following:
 - The sensitivity of the algorithm in terms of $\tau$ is not explored.
 - Wasserstein distance is mentioned in Section 2, but the proposed approach applies to Bregman divergences. Does the proposed method also work for Wasserstein distance?
 - An honest discussion about the limitations of the method is missing.

---

> ### Author Rebuttal · Authors · 2026-03-30
>
> We thank the reviewer for their thoughtful assessment.
>
> **1. Experimental scope and diversity.**
> Our experiments go beyond label shift and cover multiple distinct shifts:
>
> - **Label shift / long-tail imbalance:** CIFAR-10-LT
> - **Group shift with spurious correlations:** Waterbirds, where train/test distributions differ in label-background correlation (land vs. water)
> - **Domain generalization / covariate shift:** Bike Sharing and Concrete Strength, where we train on a subset of environments and test on another (e.g., train on three seasons, test on the fourth in Bike Sharing)
>
> These settings span classification and regression, synthetic and real-world data, and shifts beyond label reweighting.
>
> To further strengthen this point, we ran an additional domain generalization experiment on the DomainBed TerraIncognita benchmark using a ViT-style DeiT-S/16 backbone pretrained on ImageNet-1k and fine-tuned for 10 epochs. We intentionally do not use location labels during training, so the comparison is restricted to methods applicable without environment annotations. On this challenging real-world camera-trap benchmark, IRS achieves **51.81%** test accuracy, outperforming **SAM (50.97%)**, **ERM (47.92%)**, and **CVaR-DRO (21.34%)**. These results indicate that IRS remains effective beyond label shift, including challenging covariate/domain shifts under a stronger architecture and a short training budget.
>
> **2. Applicability to covariate / input shifts.**
> Our experiments already include multiple settings with input (covariate) shifts and domain generalization: (i) Waterbirds shifts the background distribution via spurious label-environment correlations; (ii) Bike Sharing and Concrete Strength train on some environments and test on unseen ones; and (iii) DomainBed TerraIncognita further evaluates real-world domain shifts.
>
> Across these settings, IRS consistently improves worst-group/worst-case performance over ERM and SAM, indicating effectiveness beyond label shift.
>
> From a modeling perspective, IRS controls sensitivity to reweighting over the observed support or structured groups. While this does not explicitly model arbitrary support-changing shifts, our results suggest this is sufficient in many practical covariate-shift settings. We will clarify this distinction in the final version.
>
> **3. KL-RS comparison.**
> KL-RS and IRS optimize the same RS objective, but differ fundamentally in optimization. KL-RS relies on outer-loop procedures (e.g., retraining for different parameter values), whereas IRS directly optimizes fragility in a single gradient-based run.
>
> In our experiments, methods are compared under a fixed computational budget. KL-RS must distribute this budget across multiple runs, while IRS uses it in a single optimization. This difference is inherent to the methods, not an artifact of the evaluation. Letting KL-RS fully converge at each outer iteration would substantially increase its cost; IRS is designed to avoid this overhead.
>
> **4. Choice of groups ($G$).**
> The grouping is dataset-dependent and follows standard practice: for CIFAR-10-LT, groups correspond to classes; for Waterbirds, to attribute combinations (background–label pairs). We will clarify this in the paper.
>
> **5. Choice of $\tau$.**
> We agree that $\tau$ is important. In our experiments, $\tau$ is selected based on a target in-distribution performance level, as discussed in Appendix B. This gives an interpretable control of the nominal accuracy/robustness tradeoff.
>
> Unlike standard hyperparameters, $\tau$ is not intended to be tuned via validation on the training distribution, since that would bias the solution toward ERM. Instead, it encodes a desired performance threshold based on domain requirements.
>
> A more systematic empirical study of $\tau$ would be valuable, and we will add discussion clarifying its role and practical selection.
>
> **6. Divergence choice.**
> Our formulation is general, but the 1D reduction used in IRS requires more than Bregmanity, specifically invertibility of the mapping $T$ defined in the paper (satisfied by Legendre-type $\phi$).
>
> In our experiments we focus on KL divergence, which satisfies these conditions and is standard on the probability simplex. We will clarify this and explicitly note that the current analysis does not cover Wasserstein distances.
>
> **7. Limitations.**
> We agree that a clearer discussion of limitations would strengthen the paper. IRS is designed for settings where the deployment distribution is uncertain and can be modeled through reweighting over the observed support or structured groups. When the target distribution is known, directly optimizing for that distribution may be more appropriate.
>
> Additionally, our current formulation does not explicitly model support-changing covariate shifts. While our experiments show strong performance in several domain generalization settings, extending RS to more general distribution shifts is an important direction for future work.

---

> > ### Author Rebuttal · Reviewer_HVHy · 2026-04-04
> >
> > Thank you to the authors for their response. This addresses all concerns raised in my review except the comparison with KL-RS. The fact that the methods are compared with a fixed computational budget is not explicitly stated in the paper (or I didn't find it). More details and the exact evaluation protocol need to be stated.
> >
> > More generally, effectiveness, efficiency and effectiveness achieved under computational constraints are three different evaluation objectives. Please clearly specify which ones are followed in the paper.

---

> > > ### Author Response · Authors · 2026-04-07
> > >
> > > We thank the reviewer for the clarification request and apologize for the confusion caused by our earlier phrasing. Our previous response was not precise enough, and we appreciate the opportunity to clarify this point.
> > >
> > > Our primary evaluation objective is **effectiveness under a matched training protocol**, not equal computational budget. Within each benchmark, all methods use the same backbone, data pipeline, evaluation protocol, and training horizon (number of epochs), consistent with standard setups used in prior work [1,2,3]. We therefore match the **training horizon** rather than total compute. Since computation within an epoch is method-dependent, this evaluates effectiveness under identical training schedules.
> > >
> > > KL-RS and IRS optimize the same robust satisficing objective but differ fundamentally in optimization procedure. KL-RS performs a bisection search over feasible fragility values, retraining for each candidate. Under a fixed epoch budget, this splits the training horizon across feasibility checks. In contrast, IRS directly minimizes fragility with gradient-based updates and uses the full training horizon along a single trajectory. Consequently, KL-RS does not receive a full training run for each candidate value, while IRS and other baselines use the entire horizon. We will clarify this in the revision. During the discussion period, we also ran additional experiments to analyze the learning behaviors of IRS and KL-RS. More extensive versions of this learning-curve analysis will be included in the appendix of the final paper.
> > >
> > > The weaker performance of KL-RS is not primarily due to the epoch budget. When a feasibility check fails, the bisection increases $\kappa$. Since $\kappa$ divides the KL-RS gradient (Eq. 12 in [4]), larger values reduce gradient magnitude, slowing learning. This makes subsequent checks more likely to fail, further increasing $\kappa$ and creating a feedback loop where gradients progressively diminish. In this regime, additional epochs do not improve performance. Abrupt changes in $\kappa$ during bisection also modify the effective loss landscape and alter the learning trajectory. In contrast, IRS follows a single gradient-based path, yielding smoother and more stable learning curves, as seen in the results given below.
> > >
> > > [1] Arjovsky et al., Invariant risk minimization, 2019
> > > [2] Sagawa et al., Distributionally robust neural networks, 2020
> > > [3] Foret et al., Sharpness-aware minimization, 2021
> > > [4] Yan et. al., KL-RS, 2024
> > >
> > > In these experiments, we used the same datasets (Bike and Concrete) and an MLP with two hidden layers of size 256 with ReLU activation. Anonymous link to plots and summary table:
> > >
> > > https://anonymous.4open.science/r/ICML-2026-Conference-17680-Figures-AA76/
> > >
> > > **IRS vs KL-RS: Train/Test MSE and fragility every 25 epochs (mean ± std across seeds)**
> > >
> > > | Dataset | Epoch | IRS Train MSE | IRS Test MSE | $\kappa$ (IRS) | KL-RS Train MSE | KL-RS Test MSE | $\kappa$ (KL-RS) |
> > > |--------|------|---------------|--------------|----|-----------------|----------------|----|
> > > | **Bike** | 25  | 0.29 ± 0.00 | 0.61 ± 0.01 | 14.94 ± 1.69 | 0.28 ± 0.04 | 0.60 ± 0.08 | 1.00 ± 0.00 |
> > > |  | 50  | 0.18 ± 0.00 | 0.48 ± 0.01 | 8.44 ± 1.24 | 0.15 ± 0.01 | 0.43 ± 0.05 | 1.00 ± 0.00 |
> > > |  | 75  | 0.14 ± 0.00 | 0.45 ± 0.02 | 5.99 ± 0.45 | 0.12 ± 0.01 | 0.42 ± 0.05 | 2.00 ± 0.00 |
> > > |  | 100 | 0.12 ± 0.00 | 0.43 ± 0.01 | 4.64 ± 0.45 | 0.10 ± 0.01 | 0.44 ± 0.06 | 2.00 ± 0.00 |
> > > |  | 125 | 0.11 ± 0.00 | 0.44 ± 0.03 | 4.23 ± 0.54 | 0.09 ± 0.01 | 0.46 ± 0.07 | 3.50 ± 1.05 |
> > > |  | 150 | 0.11 ± 0.01 | 0.44 ± 0.02 | 3.27 ± 0.91 | 0.08 ± 0.01 | 0.48 ± 0.10 | 3.50 ± 1.05 |
> > > |  | 175 | 0.10 ± 0.00 | 0.43 ± 0.01 | 1.32 ± 0.80 | 0.08 ± 0.00 | 0.52 ± 0.13 | 2.65 ± 0.74 |
> > > |  | 200 | 0.16 ± 0.14 | 0.46 ± 0.08 | 0.76 ± 0.68 | 0.07 ± 0.01 | 0.55 ± 0.17 | 2.65 ± 0.74 |
> > > |  | 225 | 0.09 ± 0.01 | 0.43 ± 0.02 | 0.70 ± 0.87 | 0.08 ± 0.05 | 0.60 ± 0.19 | 2.23 ± 0.58 |
> > > |  | 250 | 0.10 ± 0.02 | 0.43 ± 0.01 | 0.43 ± 0.36 | 0.17 ± 0.33 | 0.71 ± 0.27 | 2.23 ± 0.58 |
> > > | **Concrete** | 25  | 0.62 ± 0.02 | 1.32 ± 0.06 | 0.56 ± 0.07 | 0.29 ± 0.00 | 1.31 ± 0.03 | 1.00 ± 0.00 |
> > > |  | 50  | 0.58 ± 0.02 | 1.23 ± 0.05 | 0.35 ± 0.06 | 0.27 ± 0.00 | 1.26 ± 0.03 | 1.00 ± 0.00 |
> > > |  | 75  | 0.53 ± 0.03 | 1.17 ± 0.04 | 0.18 ± 0.09 | 0.26 ± 0.00 | 1.31 ± 0.11 | 0.50 ± 0.00 |
> > > |  | 100 | 0.49 ± 0.04 | 1.12 ± 0.04 | 0.04 ± 0.11 | 0.25 ± 0.01 | 1.22 ± 0.08 | 0.50 ± 0.00 |
> > > |  | 125 | 0.45 ± 0.03 | 1.10 ± 0.04 | -0.07 ± 0.11 | 0.39 ± 0.08 | 1.51 ± 0.47 | 0.25 ± 0.00 |
> > > |  | 150 | 0.42 ± 0.03 | 1.07 ± 0.03 | -0.17 ± 0.11 | 0.33 ± 0.02 | 1.39 ± 0.17 | 0.25 ± 0.00 |
> > > |  | 175 | 0.41 ± 0.02 | 1.06 ± 0.03 | -0.23 ± 0.09 | 0.33 ± 0.02 | 1.39 ± 0.15 | 0.38 ± 0.00 |
> > > |  | 200 | 0.40 ± 0.01 | 1.07 ± 0.03 | -0.27 ± 0.11 | 0.32 ± 0.02 | 1.34 ± 0.09 | 0.38 ± 0.00 |
> > > |  | 225 | 0.39 ± 0.01 | 1.09 ± 0.04 | -0.29 ± 0.08 | 0.32 ± 0.02 | 1.32 ± 0.05 | 0.35 ± 0.06 |
> > > |  | 250 | 0.38 ± 0.02 | 1.09 ± 0.04 | -0.34 ± 0.09 | 0.31 ± 0.02 | 1.32 ± 0.06 | 0.35 ± 0.06 |

---

### Official Review · Reviewer_ZFvs · 2026-03-19

**Soundness:** 3
**Presentation:** 2
**Significance:** 3
**Originality:** 4
**Overall Recommendation:** 5
**Confidence:** 3

**Summary:**

This paper introduces iterative robust satsficing (IRS), which is an algorithm for training neural networks under the robust satificing objective, which aims to find parameters that have bounded training error and minimize fragility. IRS optimizes robust satisficing in one training run and has similar cost to standard SGD, but the paper shows that IRS improves robustness and worst group performance while preserving in-distrubtion accuracy on a number of snythetic and real distribution shifts.

**Compliance With Llm Reviewing Policy:**

Affirmed.

**Final Justification:**

The rebuttal adressed my main concern, especially weakness #1 and #2.

**Key Questions For Authors:**

Questions

1. If I understand correctly, when minizing the fragility, we minimizes over all the test distribution d. So for different d(.,.), different test distributions will perform better as we decrease the fragility. Is there a way to qualitatively understand what test distributions get better guarantees for a given fragility level?
2. In practice, we often do not want to generalize to all possible distributions but rather only to one specific training distribution. Could we defined fragility with respect to only one training distribution d? Could we get stronger results in that case? What are the tradeoffs of specializing fragility to only one test distribution?
3. Why is the uncertainty set covering all distributions on $\{z_1,..,z_n\}$?  Could having further restrictions on $p_i$ (i.e. all $p_i$ are lower bounded) make the results stronger?
4. Why is it sufficient to only use small batches?
5. In line 127, the authors write that “$\tau$ should increase at most linearly and in the smallest possible way in d(P,P)”. What does this mean exactly? Why should it be only linear?
6. What are other ways of optimizing RS objectives besides KL-RS?
7. Are there any settings that you tried IRS where it did not work well?

**Limitations:**

Yes.

**Strengths And Weaknesses:**

Strengths
1. Training networks to perform well under distribution shift is an important problem in modern machine learning and IRS develops further a promising line work of robust satisficing, which avoids problems encountered by DRO and invariance based approaches, such as worst case nature, and need for group annotations.
2. IRS is a simple, inexpensive, and effective way to optimize RS objectives for NN and seems to perform well in practice. Empirical results are especially encouraging, and it would be interesting to further understand why IRS works.

Weaknesses
1. If I understand correctly, the IRS framework, at least as presented in the paper, only works for distribution shifts that reweigh the points in the training data, but it is not applicable to the case when the support of the distribution changes. To me his is a more interesting, and realistic case of distribution shift. The paper should at least discuss these limitations clearly.
2. Apart from brief discussion  when introducing RS, the paper is quite unclear on why RS  and IRS actually work. In principle, it’s clear that having small $d(P_d,P_D)$ and small loss on the training distribution would guarantee small error on test distribution $d$, it’s not immediate that IRS actually ensures this. Are there any empirical or theoretical evidence for this? There might be “generalization error” type bounds for $d(.,.)$ that I am missing. I would like to see more discussion on how and why IRS works.
3. The set of conditions for which the IRS approach is applicable are not clearly stated in one place. We start with general $d(.,.)$ but then later restrict to only Bregman divergence. Then we implicitly assume that mini-batching for calculating fragility works, which probably further restricts the type of $d(.,.)$ allowed.

---

> ### Author Rebuttal · Authors · 2026-03-30
>
> Dear Reviewer ZFvs,
>
> **1. Distribution support.** IRS models robustness over a fixed reference support, or over structured groupings such as classes/environments. This covers label shift and group shift. We agree that arbitrary support-changing covariate shifts are not directly covered and will state this limitation explicitly. Waterbirds and long-tailed classification are naturally described by such reweighing shifts, which is exactly the regime IRS targets. Extending RS to support-changing shifts is important future work. One possible extension is augmenting the support with synthetic samples.
>
> **2. Why RS works.** RS minimizes fragility, the smallest slope $k$ such that
> $$
> L_P(\theta) \le \tau + k\, d(P,\hat P_D)
> $$
> for all $P$ in the chosen uncertainty family. Thus IRS directly optimizes a uniform upper bound on degradation under shift. This is not a classical i.i.d. bound: RS explicitly models distributional uncertainty and finds the smallest linear envelope upper-bounding loss over the specified family. The resulting $\kappa^* $ guarantees that, within this family, degradation is at most linear in divergence, at the minimal achievable rate. This differs from DRO, which minimizes worst-case loss at a fixed radius; RS controls the degradation rate across radii.
>
> **3. Set of conditions.** We agree the assumptions should be stated more explicitly. The framework has different assumptions at different levels: **objective:** RS is defined for a general divergence/distance $d$ over a chosen family of candidate distributions; **algorithmic:** the 1D trajectory used by IRS is derived for Bregman divergences with Legendre-type $\phi$ (e.g., KL, squared Euclidean), reducing the inner maximization to a scalar search; **implementation:** mini-batch IRS is a stochastic approximation to the empirical fragility objective, analogous to SGD. **structured variants:** Class-wise and group-wise IRS are used when the uncertainty family is defined over classes or groups rather than individual samples. This captures important practical shift settings such as label shift, and group shifts.
>
> **Q1. Which test distributions benefit?** All distributions in the chosen family benefit, since minimizing fragility reduces the slope in the bound above. Distributions closer to $\hat P_D$ receive tighter guarantees. If $P^* $ attains fragility, every distribution closer to $\hat P_D$ than $P^* $ has a strictly smaller upper bound on loss. The choice of $d$ determines which shifts are emphasized.
>
> **Q2. Specializing to a single target distribution.** If $P_{\text{test}}$ is known, directly minimizing $L_{P_{\text{test}}}(\theta)$ should give the best performance on that target. RS is for uncertain or varying deployment conditions: optimizing for one target can over-specialize, whereas RS gives a uniform guarantee over a family of plausible distributions. The tradeoff is specialization versus robustness.
>
> **Q3. Choice of distributional family.** The general formulation uses distributions supported on the empirical data, capturing broad reweighting-based shifts without assuming prior knowledge of shift structure. Adding structure can yield sharper guarantees; this is exactly the role of class-wise/group-wise IRS, used in CIFAR-10-LT and Waterbirds. Unlike DRO, far distributions are penalized by distance, so enlarging the family does not overly emphasize extreme cases.
>
> **Q4. Why are small mini-batches sufficient?** In principle the exact fragility objective is full-batch. In practice, mini-batch IRS is a stochastic approximation, analogous to mini-batch SGD: each step computes losses on a mini-batch and solves the corresponding inner reweighting problem on that batch, yielding a stochastic surrogate of the fragility gradient. Full-batch computation is much more expensive; empirically, mini-batch IRS is effective.
>
> **Q5. “Increase at most linearly.”** This is by definition, not an empirical assumption: RS seeks the smallest slope $k$ such that the bound above holds for all candidate distributions. “In the smallest possible way” means minimizing this slope. Linearity yields a simple uniform certificate across divergence levels; nonlinear envelopes would define a different objective.
>
> **Q6. Other ways of optimizing RS objectives.** RS-type objectives have been studied in operations research and in sequential decision-making settings. However, to the best of our knowledge, prior work has not developed a direct single-run gradient-based method for training modern neural networks under the RS objective.
>
> **Q7. When may IRS be less suitable?** IRS is most useful when deployment shift is uncertain and robustness over a family of plausible shifts is the main goal. If the target distribution is known, directly optimizing for it is usually more appropriate. IRS is also less directly suited to shifts outside the chosen uncertainty family, such as arbitrary support-changing covariate shifts under the fixed-support formulation.

---

> > ### Author Rebuttal · Reviewer_ZFvs · 2026-04-03
> >
> > I thank the authors for answering all my questions. I will increase my score.

---

### Decision · Program_Chairs · 2026-04-30

**Decision:**

Accept (regular)

**Comment:**

The paper proposes propose Iterative Robust Satisficing, a gradient-based procedure for training neural networks under the RS objective.  Reviewers found the idea meaningful, particularly the fragility-based perspective as an alternative to standard DRO, and viewed the algorithmic reduction to an efficient single-run procedure as elegant. The empirical results were also generally seen as strong, and the application to computer vision tasks on CIFAR-10-LT exceeds traditional works in robust optimization. The reviews are consistently positive after rebuttal, and I therefore support acceptance.